



# A Radar Reflectivity Operator with Ice-Phase Hydrometeors for Variational Data Assimilation (RadZIceVar v1.0) and Its Evaluation with Real Radar Data

Shizhang Wang[1,2], Zhiquan Liu[2]

[1]Collaborative Innovation Center on Forecast and Evaluation of Meteorological Disasters, Key Laboratory of Meteorological Disaster of Ministry of Education, Nanjing University of Information Science and Technology, Nanjing, 210044, China
[2] National Center of Atmospheric Research, Boulder, 80301, USA

*Correspondence to*: Zhiquan Liu (liuz@ucar.edu)

**Abstract.** A reflectivity forward operator and its associated tangent linear and adjoint operators (together named RadZIceVar) were developed for variational data assimilation (DA). RadZIceVar can analyze both rainwater and ice-phase species (snow and graupel) by directly assimilating radar reflectivity observations. The results of three-dimensional variational (3DVAR) DA experiments with a 3 km grid mesh setting of the Weather Research and Forecasting (WRF) model showed that RadZIceVar was effective at producing an analysis of reflectivity pattern and intensity similar to the observed data. Two to three outer loops with 50-100 iterations in each loop were needed to obtain a converged 3D analysis of rainwater, snow, and graupel, including the melting layers with mixed-phase hydrometeors. The deficiencies in the analysis using this operator could be caused by the poor quality of the background fields and the use of the static background error covariance, and these issues can be partially resolved by using radar-retrieved hydrometeors in a preprocessing step and tuning the spatial correlation length scales of the background errors. The direct radar reflectivity assimilation using RadZIceVar also improved the short-term (2 h-5 h) precipitation forecasts compared to those of the experiment without DA.

## 1 Introduction

Over the past several decades, radar reflectivity observations have been used in many data assimilation (DA) studies (Borderies et al., 2018; Caumont et al., 2010; Gao and Stensrud, 2012; Hu et al., 2006; Jung et al., 2010; Jung et al., 2008; Liu et al., 2018; Putnam et al., 2014; Snook et al., 2012, 2015; Sun and Crook, 1997; Sun and Wang, 2013; Tong and Xue, 2005; Wang et al., 2013b; Wang and Wang, 2017; Wattrelot et al., 2014; Xiao et al., 2007; Xue et al., 2006) and have demonstrated that assimilating this radar reflectivity improves the initial conditions of the convective scale and benefits the subsequent forecasts. To assimilate the reflectivity, it is necessary to transform the model's prognostic variables (e.g., rainwater, snow, and graupel) to the observed radar reflectivity. To perform this transformation, early studies (e.g., Sun and Crook, 1997; Xiao et al., 2007) used the Marshal–Palmer distribution of raindrop size (Z-R relationship). However, this relationship is only valid in precipitation areas without ice-phase species; thus, its applications (e.g., Schwitalla and



Wulfmeyer, 2018) are often limited in layers lower than 4 km or 8 km above ground level (AGL). To overcome this deficiency, more comprehensive operators that involve snow and graupel have been developed (Gao and Stensrud, 2012; Tong and Xue, 2005). Several studies (e.g., Gao and Stensrud, 2012; Wang and Wang, 2017) have demonstrated that involving these ice species in the reflectivity operator improves the analysis of hydrometeors in terms of their spatial

distribution, especially in the vertical direction.

Although these operators have been successfully applied in many convective-scale DA studies, they were developed for a specific band of radar (e.g., S-band; Gao and Stensrud, 2012; Sun and Crook, 1997) and specific microphysics characteristics (e.g., with a fixed intercept parameter; Sun and Crook, 1997). However, mixed-phase species such as wet snow and wet graupel have not been considered in these operators. Recently, the contributions from mixed-phase species have been studied

(Jung et al., 2008, hereafter, J08; Posselt et al., 2015). To compute the mixed-phase species' contributions, J08 proposed an operator that was based on the expressions given by Zhang et al. (2001). Their expressions were derived according to the scattering amplitudes that were estimated through the T-matrix method and the Rayleigh scattering approximation. For computational efficiency, these expressions were rewritten in the polynomial form that was only valid for S-band radar and the conditions that the Rayleigh assumption was satisfied. Later, a more general and exact operator that fully used the T-

matrix scattering method was proposed (Jung et al., 2010). This operator was given as the integral of the complex backscattering amplitudes over the size distribution of the precipitation particles (i.e., rainwater, snow, and graupel). In addition to these operators, several complex reflectivity operators in the integral form have also been proposed (Borderies et al., 2018; Caumont et al., 2006; Pfeifer et al., 2008; Ryzhkov et al., 2011; Wattrelot et al., 2014). Some were designed for a specific band of radar (e.g., W-band; Borderies et al., 2018), whereas some were designed for the bin microphysics scheme

(e.g., Ryzhkov et al., 2011).

Currently, two approaches utilize these reflectivity operators. One is the variational DA method (Caumont et al., 2010; Gao and Stensrud, 2012; Hu et al., 2006; Sun and Crook, 1997; Sun and Wang, 2013; Wang et al., 2013b; Wattrelot et al., 2014; Xiao et al., 2007), and the other is the ensemble Kalman filter method (EnKF; Dawson et al., 2010; Jung et al., 2010; Jung et al., 2008; Putnam et al., 2014; Snook et al., 2011, 2015; Tong and Xue, 2005; Xue et al., 2006). The variational method

requires the tangent linear (TL) and adjoint (AD) operators, which are not required by the EnKF (Evensen, 2003). Therefore, complex operators such as those proposed by J08 are often employed in EnKF DA applications. For the variational method, a common approach to avoid using the TL/AD operators is to assimilate the reflectivity-retrieved hydrometeor profiles (Caumont et al., 2010; Wang et al., 2013a; Wattrelot et al., 2014); an alternative is to use the reflectivity as an additional control variable with the ensemble-variational DA approach (Wang and Wang, 2017).

Despite the difficulty, many efforts have focused on reflectivity assimilation with the TL/AD operators (Gao and Stensrud, 2012; Kawabata et al., 2018; Liu et al., 2018; Xiao et al., 2007), and reasonable results have been obtained in terms of hydrometeor analysis and precipitation forecasts. However, none of these studies employed operators as complex as those proposed by J08. Kawabata et al. (2018) adopted the expressions of Zhang et al. (2001) and developed the TL/AD operator for C-band radar but without taking into account the contributions from ice-phase species.



The main purpose of this study is to develop a TL/AD operator based on Jung et al. (2008) with the contributions of ice-phase precipitation and apply it in a variational DA framework. For convenience, the operator implemented in this study is called RadZIceVar to represent that it was developed for variational DA and contains ice-phase species. The original J08 operator is called J08orig. The reminder of this paper is organized as follows. In Section 2, the J08 operator is reviewed, and

its TL and AD operators are derived. The experimental design is given in Section 3, and the new operators are verified in Section 4. The performance of RadZIceVar is discussed in Section 5, and the conclusions are presented in Section 6.

## 2 Reflectivity operator

### 2.1 Review of the J08 operator

The radar-observed reflectivity, $Z$, is given in logarithmic form as

$$Z = 10\log_{10} Z_{\mathrm{e}}, \tag{1}$$

where $Z_{\mathrm{e}}$ is the equivalent reflectivity factor, which is the sum of the contributions from pure rainwater ($Z_{\mathrm{r}}$), dry snow ($Z_{\mathrm{ds}}$), dry graupel ($Z_{\mathrm{dg}}$), wet snow ($Z_{\mathrm{ws}}$), and wet graupel ($Z_{\mathrm{wg}}$) as follows:

$$Z_{\mathrm{e}} = Z_{\mathrm{r}} + Z_{\mathrm{ds}} + Z_{\mathrm{dg}} + Z_{\mathrm{ws}} + Z_{\mathrm{wg}}. \tag{2}$$

To compute Eq. (2), it is necessary to use the mixing ratios of mixed-phase species (wet snow and wet graupel). However,

many widely used microphysics schemes, such as the Lin, WSM6, and Morrison schemes, do not predict or diagnose the mixed-phase species; thus, the amount of rainwater in wet snow or graupel cannot be directly extracted from the model output. To solve this issue, J08 modeled the rain-snow (rain-graupel) mixture using a fraction that is given by

$$F = [\min(q_{\mathrm{r}} / q_{\mathrm{x}}, q_{\mathrm{x}} / q_{\mathrm{r}})]^{0.3} F_{\max}, \tag{3}$$

where $F_{\max}$ is the maximum fraction, which is 0.5 (0.3) for rain-graupel (rain-snow) mixtures; $q_{\mathrm{r}}$ is the mixing ratio of

rainwater; and $q_{\mathrm{x}}$ is the general form of the mixing ratio of ice-phase species. The subscript "x" can be either "s" for snow or "g" for graupel. With this fraction, the mixing ratios of pure rainwater, dry snow, dry graupel and mixed-phase species are given by

$$
\begin{aligned}
q_{\mathrm{pr}} &= (1 - F_{\mathrm{ws}} - F_{\mathrm{wg}})q_{\mathrm{r}} \\
q_{\mathrm{ds}} &= (1 - F_{\mathrm{ws}})q_{\mathrm{s}} \\
q_{\mathrm{dg}} &= (1 - F_{\mathrm{wg}})q_{\mathrm{g}} \\
q_{\mathrm{ws}} &= F_{\mathrm{ws}}(q_{\mathrm{s}} + q_{\mathrm{r}}) \\
q_{\mathrm{wg}} &= F_{\mathrm{wg}}(q_{\mathrm{g}} + q_{\mathrm{r}})
\end{aligned}
\tag{4}
$$

where the subscripts "ws" and "wg" are added to $F$ to represent the fractions of wet snow and wet graupel, respectively, and

the subscripts "pr", "ds", and "dg" represent pure water, dry snow, and dry graupel, respectively. The mixed-phase density, $\rho_{\mathrm{wx}}$, is not a constant and is parameterized by





$$\rho_{wx} = (1 - f_{wx}^2)\rho_x + f_{wx}^2\rho_r \qquad (5)$$

with

$$f_{wx} = \frac{q_r}{q_r + q_x}. \qquad (6)$$

The subscript "x" in $\rho_{wx}$, $\rho_x$, and $f_{wx}$ represents either snow (s) or graupel (g), and $f_{wx}$ is called the water fraction.

### 2.1.1 Contribution from rainwater

In accordance with J08, all of the contributions are computed by integrations over the drop size distribution (DSD) weighted by the scatter cross section determined by the density, shape, and DSD. The DSD is modeled by an exponential distribution. After performing the integration, the contribution from pure rainwater, $Z_r$, is written in a simple form (Zhang et al., 2001; Posselt et al., 2015; Kawabata et al., 2018) as follows:

$$Z_r = \frac{4\lambda^4\alpha_{ra}^2 N_{0r}}{\pi^4 |K_w|^2}\Lambda_r^{-(2\beta_{ra}+1)}\Gamma(2\beta_{ra}+1), \qquad (7)$$

where $\lambda$ is the wavelength of the radar, which is 107 mm for S-band radar, and $N_{0r}$ is the intercept parameter of rainwater, which is $8\times10^6$ m$^{-4}$ in this study. A fixed $N_{0r}$ value is only available for a single moment microphysics scheme; for a two-moment scheme, this value should be determined using the predicted number concentration. $K_w$ is the dielectric factor for rainwater and is equal to 0.93, and $\alpha_{ra}$ and $\beta_{ra}$ are $4.28\times10^{-4}$ and 3.04, respectively. The complete gamma function is written as $\Gamma(\ldots)$. The slope parameter of rain, $\Lambda_r$, is

$$\Lambda_r = (\frac{\pi\rho_r N_{0r}}{\rho_a q_{pr}})^{\frac{1}{4}}, \qquad (8)$$

where $\rho_r = 1000$ kg m$^{-3}$ is the rain density, $q_{pr}$ is given by Eq. (4), and $\rho_a$ is the density of air. By substituting Eq. (8) and the constant parameters into Eq. (7), we can rewrite Eq. (7) as a function of $q_{pr}$ as follows:

$$Z_r(q_{pr}) = P_r(q_{pr})^{1.77}, \qquad (9)$$

where

$$P_r = \frac{4\lambda^4\alpha_{ra}^2}{\pi^4 |K_w|^2}(\frac{\pi\rho_r}{\rho_a})^{-\frac{2\beta_{ra}+1}{4}}(N_{0r})^{1-\frac{2\beta_{ra}+1}{4}}\Gamma(2\beta_{ra}+1). \qquad (10)$$

The value of $P_r$ is approximately $4.8\times10^9$ with an air density of 1.0 kg m$^{-3}$. This value has the same magnitude as those proposed by Sun and Crook (1997) and Gao and Stensrud (2012).

### 2.1.2 Contribution from dry snow/graupel

The contributions from both dry and mixed-phase ice species after integration have the same form but differ in their coefficients. For dry ice species, the contribution is given by





$$Z_{\mathrm{dx}} = \frac{4\Gamma(7)\lambda^4 N_{0x}}{\pi^4 \left|K_w\right|^2} \Lambda_{\mathrm{dx}}^{-7}(A\alpha_{\mathrm{dxa}}^2 + B\alpha_{\mathrm{dxb}}^2 + 2C\alpha_{\mathrm{dxa}}\alpha_{\mathrm{dxb}}), \tag{11}$$

where the subscript $x$ represents either snow (s) or graupel (g). The intercept parameters of these species are denoted by $N_{0x}$, the values of which are $3\times10^6$ m$^{-4}$ and $4\times10^5$ m$^{-4}$ for snow and graupel, respectively. The slope parameter in Eq. (11) for either dry snow or dry graupel is written as

$$\Lambda_{\mathrm{dx}} = (\frac{\pi\rho_x N_{0x}}{\rho_a q_{\mathrm{dx}}})^{\frac{1}{4}}. \tag{12}$$

The parameters $A$, $B$, and $C$ in Eq. (11) are functions of the mean ($\bar{\phi}$) and the standard deviation ($\sigma$) of the canting angle and are given by

$$A = \frac{1}{8}(3 + 4\cos 2\bar{\phi}e^{-2\sigma^2} + \cos 4\bar{\phi}e^{-8\sigma^2})$$
$$B = \frac{1}{8}(3 - 4\cos 2\bar{\phi}e^{-2\sigma^2} + \cos 4\bar{\phi}e^{-8\sigma^2}) \cdot \tag{13}$$
$$C = \frac{1}{8}(1 - \cos 4\bar{\phi}e^{-8\sigma^2})$$

According to J08, $\bar{\phi}$ is zero for all hydrometeors, and $\sigma$ is different for snow (20°) and hail (60°). In Eq. (11), these functions are multiplied by coefficients ($\alpha_{\mathrm{dxa}}$ and $\alpha_{\mathrm{dxb}}$) that describe the backscattering amplitudes. For dry ice-phase species, these coefficients are precalculated constants and are listed in Table 1.

For brevity, Eq. (11) is rewritten as a function of $q_{\mathrm{dx}}$ for dry snow and dry graupel and is given by

$$Z_{\mathrm{dx}}(q_{\mathrm{dx}}) = P_{\mathrm{dx}}q_{\mathrm{dx}}^{1.75}, \tag{14}$$

where

$$P_{\mathrm{dx}} = \frac{4\Gamma(7)\lambda^4 N_{0x}^{-0.75}}{\pi^4 \left|K_w\right|^2}(\frac{\pi\rho_x}{\rho_a})^{-1.75}(A\alpha_{\mathrm{dxa}}^2 + B\alpha_{\mathrm{dxb}}^2 + 2C\alpha_{\mathrm{dxa}}\alpha_{\mathrm{dxb}}) \cdot \tag{15}$$

### 2.1.3 Contribution from wet snow/graupel

The equation for the contribution from mixed-phase species has the same form as Eq. (11) except that the subscript "d" is replaced by "w" to represent wet species. The slope parameter for mixed-phase species also has the same form as Eq. (12) except that the subscript "d" is replaced by "w", and $\rho_{wx}$ substitutes for $\rho_x$. For wet species, $\sigma$ in A, B, and C is a function of $f_w$ and $q_{wg}$. Additional details are given in Section 3c of J08. The coefficients that are multiplied by $A$, $B$, and $C$ are functions of $f_w$ and are written as





$$\alpha_{\mathrm{wxa}} = \varepsilon_{\mathrm{x}} \sum_{k=0}^{n} P_{\mathrm{wxa}k} f_{wx}^{k} , \tag{16}$$

$$\alpha_{\mathrm{wxb}} = \varepsilon_{\mathrm{x}} \sum_{k=0}^{n} P_{\mathrm{wxb}k} f_{wx}^{k}$$

where "x" is "s" ($g$) for snow (graupel), $\varepsilon_{\mathrm{x}}$ is $10^{-4}$ ($10^{-3}$) for snow (graupel), $P_{\mathrm{wxa}k}$ and $P_{\mathrm{wxb}k}$ are precalculated constants for S-band radar, and the superscript $k$ denotes the index of these constants. Based on J08, the value of $n$ is 6. The values of $P_{\mathrm{wxa}k}$ and $P_{\mathrm{wxb}k}$ are listed in Table 2.

To simplify the derivation of the TL/AD operators, we rewrite Eq. (11) as a function of the mixed-phase mixing ratio ($q_{\mathrm{wx}}$) and the water fraction ($f_{\mathrm{w}}$) as follows:

$$Z_{\mathrm{wx}}(q_{\mathrm{wx}}, f_{\mathrm{wx}}) = P_{\mathrm{wx}} q_{\mathrm{wx}}^{1.75} \varepsilon_{\mathrm{x}}^{2} \sum_{k=0}^{2n} P_{\mathrm{x}k} f_{\mathrm{wx}}^{k} , \tag{17}$$

where the coefficients $P_{\mathrm{wx}}$ and $P_{\mathrm{x}k}$ are given by

$$P_{\mathrm{wx}} = \frac{4\Gamma(7)\lambda^{4} N_{0\mathrm{x}}^{-0.75}}{\pi^{4} |K_{\mathrm{w}}|^{2}} \left(\frac{\pi\rho_{\mathrm{wx}}}{\rho_{\mathrm{a}}}\right)^{-1.75} , \tag{18}$$

and

$$P_{\mathrm{x}k} = A P_{\mathrm{Ax}k} + B P_{\mathrm{Bx}k} + 2C P_{\mathrm{Cx}k} , \tag{19}$$

respectively. $P_{\mathrm{Ax}k}$, $P_{\mathrm{Bx}k}$, and $P_{\mathrm{Cx}k}$ in Eq. (19) are given by

$$P_{\mathrm{Ax}k} = \begin{cases} \displaystyle\sum_{i=0}^{k} P_{\mathrm{wxa}i} P_{\mathrm{wxa}(k-i)} , & (k \le n) \\ \displaystyle\sum_{i=k-n}^{n} P_{\mathrm{wxa}i} P_{\mathrm{wxa}(k-i)} , & (k > n) \end{cases}$$

$$P_{\mathrm{Bx}k} = \begin{cases} \displaystyle\sum_{i=0}^{k} P_{\mathrm{wxb}i} P_{\mathrm{wxb}(k-i)} , & (k \le n) \\ \displaystyle\sum_{i=k-n}^{n} P_{\mathrm{wxb}i} P_{\mathrm{wxb}(k-i)} , & (k > n) \end{cases} , \tag{20}$$

$$P_{\mathrm{Cx}k} = \begin{cases} \displaystyle\sum_{i=0}^{k} P_{\mathrm{wxa}i} P_{\mathrm{wxb}(k-i)} , & (k \le n) \\ \displaystyle\sum_{i=k-n}^{n} P_{\mathrm{wxa}i} P_{\mathrm{wxb}(k-i)} , & (k > n) \end{cases}$$

where $P_{\mathrm{wxa}i}$ and $P_{\mathrm{wxb}i}$ are precalculated constants listed in Table 2. The subscript "x" in these constant coefficients represents

either snow (s) or graupel (g). The derivations of Eq. (17) to Eq. (20) are given in the appendix.



## 2.2 Tangent linear operator

Because RadZIceVar is highly nonlinear and complex, performing the derivation for every nonconstant variable in this operator is difficult. In addition, some components of RadZIceVar (e.g., the fraction $F$ in Eq. (3)) are discontinuous and may cause serious convergence problems in the minimization (Janisková and Lopez, 2013; Janisková et al., 1999). Although this

issue can be addressed by performing regularization for the discontinuous components in RadZIceVar, it is beyond the scope of this study. Here, we assumed that five variables in RadZIceVar are not changed in the minimization: i) the air density, ii) the fraction $F$, iii) the intercept parameter, iv) the standard deviation of the canting angle $\sigma$, and v) the density of the mixed-phase species. The air density is held constant for simplification and to focus on the impact of the changes in the hydrometeors. The fraction $F$ is assumed to remain unchanged because of its second-order discontinuity at the point that $q_x$ is

equal to $q_r$. The intercept parameter is constant because RadZIceVar was currently designed for single-moment microphysics schemes. Although multimoment microphysics schemes are more realistic because they predict the number density of hydrometeors, single-moment microphysics schemes are still widely used to compute reflectivity and the polarization variables (e.g., Posselt et al., 2015). For simplification, the standard deviation of the canting angle and the density of the mixed-phase species remain unchanged in the minimization. These assumptions are discussed in Section 4.

### 2.2.1 Linearization for rain

Considering the assumptions presented above, $P_r$ in Eq. (9) becomes a constant in the minimization. Therefore, the linearized form of Eq. (9) for $q_{pr}$ is given by

$$\delta Z_r(q_{pr}) = \frac{\partial Z_r(q_{pr})}{\partial q_{pr}} \frac{\partial q_{pr}}{\partial q_r} \delta q_r \\ = 1.77\, P_r(q_{pr})^{0.77}(1 - F_{ws} - F_{wg})\delta q_r \tag{21}$$

### 2.2.2 Linearization for dry and wet snow/graupel

For dry snow and graupel, the linearized form of Eq. (14) is given by

$$\delta Z_{dx}(q_{dx}) = \frac{\partial Z_{dx}}{\partial q_{dx}} \frac{\partial q_{dx}}{\partial q_x} \delta q_x \\ = 1.75 P_{dx} q_{dx}^{0.75}(1 - F_{wx})\delta q_x \tag{22}$$

The linearization of Eq. (17) can be categorized into two parts, which represent the variations in $Z_x$ caused by changes in $q_{wx}$ and $f_{wx}$, respectively. The linear equation of Eq. (17) is written as




$$\delta Z_{wx}(q_{wx}, f_{wx}) = \frac{\partial Z_{wx}}{\partial q_{wx}}(\frac{\partial q_{wx}}{\partial q_x}\delta q_x + \frac{\partial q_{wx}}{\partial q_r}\delta q_r) + \frac{\partial Z_{wx}}{\partial f_{wx}}(\frac{\partial f_{wx}}{\partial q_x}\delta q_x + \frac{\partial f_{wx}}{\partial q_r}\delta q_r)$$

$$= (\frac{\partial Z_{wx}}{\partial q_{wx}}\frac{\partial q_{wx}}{\partial q_r} + \frac{\partial Z_{wx}}{\partial f_{wx}}\frac{\partial f_{wx}}{\partial q_r})\delta q_r + (\frac{\partial Z_{wx}}{\partial q_{wx}}\frac{\partial q_{wx}}{\partial q_x} + \frac{\partial Z_{wx}}{\partial f_{wx}}\frac{\partial f_{wx}}{\partial q_r})\delta q_x$$

$$= \{1.75 P_{wx} q_{wx}^{0.75} F_{wx}(\varepsilon_x^2 \sum_{k=0}^{2n} P_{xk} f_{wx}^k)$$

$$+ P_{wx} q_{wx}^{1.75}[\varepsilon_x^2 \sum_{k=0}^{2n} P_{xk} k f_{wx}^k(\frac{1}{q_r} - \frac{1}{q_r + q_x})]\}\delta q_r$$

$$+ \{1.75 P_{wx} q_{wx}^{0.75} F_{wx}(\varepsilon_x^2 \sum_{k=0}^{2n} P_{xk} f_{wx}^k)$$

$$+ P_{wx} q_{wx}^{1.75}[\varepsilon_x^2 \sum_{k=0}^{2n} P_{xk} k f_{wx}^k(-\frac{1}{q_r + q_x})]\}\delta q_x$$

$$, \qquad (23)$$

where the subscript "x" represents either snow (s) or graupel (g). Using s (g) to replace the subscript "x" in Eq. (23), we can obtain the tangent linear operator of the contribution from wet snow (wet graupel).

The linearization of Eq. (1) is given by

5 $\qquad \delta Z = 10\frac{1}{Z_e \ln 10}\delta Z_e,$ $\qquad\qquad\qquad\qquad\qquad\qquad\qquad$ (24)

where $\delta Z_e$ is the sum of $\delta Z_r$, $\delta Z_{ds}$, $\delta Z_{dg}$, $\delta Z_{ws}$, and $\delta Z_{wg}$.

## 2.3 Adjoint operator

The adjoint operator is the transpose of the tangent linear operator. Because the tangent linear operator is applied to the model variables $q_r$, $q_s$, and $q_g$, the adjoint operator is written for these variables. First, the adjoint operator is written for Eq.

(24). This operator has the following form:

$$\delta Z_e^A = 10\frac{1}{Z_e \ln 10}\delta Z, \qquad\qquad\qquad\qquad\qquad\qquad\qquad (25)$$

where the superscript "A" means adjoint.

For rainwater in Eq. (21), the adjoint operator is given by

$$\delta q_r^A = \delta q_r^A + 1.77 P_r (q_{pr})^{0.77}(1 - F_{ws} - F_{wg})\delta Z_e^A. \qquad\qquad\qquad (26)$$

The parameter $q_r^A$ on the right-hand side of Eq. (26) is the accumulated $q_r^A$ before computing Eq. (26). This rule is also valid for $q_x$.

The adjoint operator of Eq. (22) is given by

$$\delta q_x^A = \delta q_x^A + 1.75 P_{dx} q_{dx}^{0.75}(1 - F_{wx})\delta Z_e^A. \qquad\qquad\qquad\qquad (27)$$





Because Eq. (23) is the derivation with respect to both $q_r$ and $q_x$, the adjoint operator of Eq. (23) contains two parts: one for rainwater and the other for ice species. For rainwater involved in Eq. (23), the adjoint operator is given by

$$\delta q_r^A = \delta q_r^A + 1.75 P_{wx} q_{wx}^{0.75} F_{wx} (\varepsilon_x^2 \sum_{k=0}^{2n} P_{xk} f_{wx}^k) \delta Z_e^A$$
$$+ P_{wx} q_{wx}^{1.75} [\varepsilon_x^2 \sum_{k=0}^{2n} P_{xk} k f_{wx}^k (\frac{1}{q_r} - \frac{1}{q_r + q_x})] \delta Z_e^A \tag{28}$$

For ice species in Eq. (23), the adjoint operator is given by

$$\delta q_x^A = \delta q_x^A + 1.75 P_{wx} q_{wx}^{0.75} F_{wx} (\varepsilon_x^2 \sum_{k=0}^{2n} P_{xk} f_{wx}^k) \delta Z_e^A$$
$$+ P_{wx} q_{wx}^{1.75} [\varepsilon_x^2 \sum_{k=0}^{2n} P_{xk} k f_{wx}^k (-\frac{1}{q_r + q_x})] \delta Z_e^A \tag{29}$$

### 2.4 Sensitivity of RadZIceVar to changes in hydrometeors

Since RadZIceVar is highly nonlinear, understanding its response to changes in $q_r$, $q_s$, and $q_g$ assists in the analysis of DA results using this operator. The response functions for Eq. (9), Eq. (14), and Eq. (17) are plotted in Fig. 1. Figure 1b is a two-dimensional plot because Eq. (17) involves two kinds of hydrometeors. Fig. 1a shows that the reflectivity changes more rapidly for all three hydrometeors when the mixing ratios are less than 0.5 g kg$^{-1}$. As the mixing ratios increase to 2.0 g kg$^{-1}$ or greater, the relationship between the reflectivity and the mixing ratio is approximately linear. This feature indicates that the reflectivity is more sensitive at small mixing ratios, and it also implies that the tangent linear approximation may give larger errors when the background reflectivity is small.

In addition, the reflectivity contribution from wet snow increases more substantially when $q_s$ or $q_r$ are in the range of $0 - 0.5$ g kg$^{-1}$. Fig. 1b shows that the reflectivity reaches 35 dBZ when both $q_s$ and $q_r$ are approximately 0.2 g kg$^{-1}$, while this reflectivity value requires $q_s$ of ~1.2 g kg$^{-1}$ for dry snow. This result is expected because many observation studies (e.g., Zhang et al., 2008) have shown that wet snow causes a bright band (large reflectivity) in the melting layer. This result also implies that the approximation error in the melting layer could be large.

## 3 Data and experimental design

### 3.1 Case review

This study focuses on a precipitation case that occurred in the northern U.S. The precipitation initiated at approximately 21Z on 1 June 2018, when convective cells formed near the border between South Dakota and Nebraska. By 00Z on 2 June 2018, these cells had developed into a linear convective system that was approximately 300 km long in the northeast-southwest direction and stretched from the middle of South Dakota to the middle of Nebraska. The top of the convective system at this





time in terms of the observed reflectivity greater than 5 dBZ reached 16 km AGL. This convective line developed further and moved to southeast Nebraska from 00Z to 04Z. During this period, a bow echo could be observed on the radar mosaic provided by the National Centers for Environmental Information (NCEI), as shown in Fig. 2. By 08Z, the convective line had moved to the northern border of Kansas, followed by a large area of stratiform clouds that covered eastern Nebraska.

The precipitation caused by this convective system lasted for nearly 20 h and ended at approximately 18Z on 2 June 2018.

## 3.2 Settings of the forecast model

Version 3.9.1 of the Weather Research and Forecasting (WRF; Skamarock et al., 2018) model was employed. All of the experiments were performed on a 450×450×42 domain centered at (41 °N, 96 °W) in eastern Nebraska (Fig. 2). The horizontal grid spacing was 3 km. The terrain-following vertical grid was employed with the model top at 50 hPa. All of the

experiments used the same physical parameterizations: no cumulus parameterization, the Thompson microphysics scheme (Thompson et al., 2004), the RRTMG longwave and shortwave radiation scheme (Iacono et al., 2008; Mlawer et al., 1997), and the Unified Noah land-surface model (Chen and Dudhia, 2001). The initial conditions and the lateral boundary conditions were generated with the Global Forecast System (GFS) data at 00Z on 2 June 2018.

## 3.3 Generation of the background error covariance

RadZIceVar was implemented in the WRF Data Assimilation (WRFDA) (Barker et al., 2012) system. To perform the variational DA with this newly developed radar operator, it is necessary to generate the background error covariance matrix with $q_r$, $q_s$, and $q_g$ as part of the control variables. The generalized software package for the background error covariance statistics (GEN_BE) developed by Descombes et al. (2015) was used. The GEN_BE package can generate the univariate background error statistics for 11 variables, including these three hydrometeors. The background error statistics were

computed using the National Meteorological Center (NMC) method (Parrish and Derber, 1992), which uses pairs of the differences between the 12 h and 24 h forecasts. A total of 27 days' forecasts from 20 May 2018 to 15 June 2018 were employed to generate the background error covariance.

The background error statistics for $q_r$, $q_s$, and $q_g$ are shown in Fig. 3. The vertical distributions of the background error's standard deviation (Fig. 3a, b, c) are consistent with those of the corresponding hydrometeor profiles: the error of the

rainwater mainly appears in the lower levels, while that of snow and graupel mainly exists in the upper levels. The graupel may fall from the upper levels into the lower levels, so the graupel error has a broad distribution. The horizontal correlation length scales of the background errors are often less than 4 grids (< 12 km), and the vertical correlation of each hydrometeor can be large at the associated precipitation levels. These spatial correlations of the hydrometeor errors determine the remote horizontal and vertical influences of the observed reflectivity.



### 3.4 Observation data and verification data

Radar data at 00Z and 01Z on 2 June 2018 were selected to evaluate the radar operator in a variational analysis framework because the convection was sufficiently deep to contain ice-phase species. To fully cover the convective system at 00Z, data from KABR, KFSD, KLNX, KOAX, KUDX, and KUEX were used, and KLNX was the closest radar from the convective

line (Fig. 2). These radars were located in Nebraska and South Dakota. The radar data were stored in Level-II format and converted to WRFDA format using a modified 88D2ARPS package, which is widely used in radar DA studies (Putnam et al., 2014; Snook et al., 2011, 2012). During this conversion, the radar data were also horizontally remapped to the model grids but remained at the radar elevations in the vertical direction; in other words, the horizontal resolution of the radar data after the conversion was consistent with that of the model. Based on other studies (e.g., Umemoto et al., 2009), the observation

error of the reflectivity was set to 1 dBZ. Our early tests using different observation errors indicated that the errors of the analysis reflectivity were comparable when using reflectivity errors ranging from 0.5 dBZ to 2.0 dBZ. For computational efficiency, we selected the remapped data every two grids in both the $x$ and $y$ directions for the DA.

The NCEP gridded stage IV (ST4) dataset (Lin and Mitchell, 2005) was used for the precipitation forecast verification. ST4 data with a horizontal resolution of 4 km were interpolated into the 3 km model grid mesh to evaluate the precipitation

prediction. At each model grid, the interpolated value was the weighted average of the ST4 data within 10 km of the grid; these data were weighted by the square of the inverse of the distance between the model grid and the ST4 data location.

### 3.5 Experimental design

As the first attempt to implement and apply RadZIceVar in WRFDA, this study focused on the quality of the analysis using the univariate 3DVAR DA method in terms of the root-mean-square error (RMSE) against the observed reflectivity and the

similarity between the observed reflectivity distributions and the analysis. The forecast performance is the secondary concern and will be explored more thoroughly in a future study with multivariate analysis using more advanced DA techniques.

All of the DA experiments analyzed only rainwater, snow, and graupel by assimilating only the radar reflectivity observations. The first DA experiment, called **Exp_ref**, is considered the benchmark experiment; it mostly used the default configurations of WRFDA-3DVAR, except the number of outer loops was set to six, and the number of maximum iterations

in each outer loop was set to 150. More outer loops is utilized to consider inaccurate background hydrometeors and the high nonlinearity of the radar operator. To determine the tradeoff between the analysis quality and computational cost, two variants of Exp_ref were conducted with 50 and 100 inner iterations. Two Exp_ref analyses at 00Z and 01Z were performed. The background for the 00Z analysis was interpolated from the GFS analysis with zeros for the hydrometeor fields, and the background for the 01Z analysis was the 1-h WRF forecast from the 00Z analysis with more realistic hydrometeor fields.

Note that TL/AD of RadZIceVar will not be able to create the change in reflectivity with hydrometeor perturbations with the zero-hydrometeor background (serves as the base state of TL/AD in the first outer loop), which is the case for the 00Z analysis. An approach to address this issue is to reset the zero background values of $q_r$, $q_s$, and $q_g$ to small values that can





range from $10^{-9}$ to $10^{-6}$ kg kg$^{-1}$ (e.g., Wang and Wang, 2017). However, this approach will result in the fraction $F$ being a constant, while J08 expected F to peak near the middle of the melting layer. Therefore, we introduced a hydrometeor preprocessing step before performing the analysis, which constructs a new background with the weighted sum of the radar-retrieved hydrometeor mixing ratios and their background counterparts. The hydrometeor retrieval followed the procedure

that is available in WRFDA, which is based on Gao and Stensrud (2012). The weight coefficients are arbitrarily set to 0.1 for the retrieval part and to 0.9 for the background. The small weight for the retrieval part was used to minimize the impact of the retrieval, mainly to ensure a nonvanishing background. The hydrometeor preprocessing was performed at both 00Z and 01Z in Exp_ref as a reference.

To observe the analysis performance with a bad background, the Exp_ref analysis at 00Z was also run with a very small

retrieval weight of $5 \times 10^{-4}$. In addition, the impact of the hydrometeor preprocessing on a relatively "good" background (01Z) was examined by comparing it with an experiment without hydrometeor preprocessing.

In several previous studies (Ban et al., 2017; Choi et al., 2017; e.g., Shen and Min, 2015), the horizontal correlation length scale factor of the background error had a large impact on the analysis. Therefore, two additional experiments, Exp_ls0.5 and Exp_ls0.125, were performed at 00Z with length scale factors of 0.5 and 0.125, respectively.

Short-term forecasts from some of these analyses and from the GFS analysis (referred to as the 'noDA' experiment) were also carried out and evaluated.

## 4 Radar operator validation

Before evaluating the performance of RadZIceVar in WRFDA-3DVAR, we first examined the consistency between J08orig and RadZIceVar. The pyCAPS-PRS v1.1 software (Dawson et al., 2014; Johnson et al., 2016; Jung et al., 2010; Jung et al.,

2008) provided by the Center for Analysis and Prediction of Storms (CAPS) was used to serve as the J08orig operator. The noDA 4-hr forecast initialized at 00Z was used as the input of the radar operators. The results, which are shown in Fig. 4, indicate that the difference between J08orig and RadZIceVar is small and acceptable. The small difference is likely caused by the subtle programming differences between the two software packages.

To verify whether RadZIceVar is sensitive to the unchanged variables during the minimization (see Section 2b), four tests

were conducted. The same noDA 4-hr forecast was used. In these tests, $q_r$, $q_s$, and $q_g$ were randomly perturbed with standard deviations proportional to their input values; in other words, a large background mixing ratio caused a large perturbation. The tests called F_unprt, rhom_unprt, and SD_unprt represent that the fraction F, the mixed form density, and the standard deviation of canting angle, respectively, were unperturbed (i.e., fixed input calculated from the forecast input). The test called all_prt denotes that all of the variables in RadZIceVar were calculated using perturbed mixing ratios. The standard

deviation of the perturbation was set to 10% of the background value; thus, the maximum perturbation could be large, as shown in Fig. 5a. Fig. 5 shows that the reflectivities computed in F_unprt, rhom_unprt, and SD_unprt do not differ significantly from those computed in all_prt. This result indicates that keeping these variables unchanged during the





minimization is an acceptable approximation. The most noticeable difference occurs in the middle layer, which is marked in red circles that plot off the diagonal line by several dBZ (Fig. 5c), but there are few of these samples.

The tangent linear operator of RadZIceVar was verified through a ratio, which is given by

$$\frac{\left|H(\mathbf{x}+\varepsilon\delta\mathbf{x})-H(\mathbf{x})\right|}{\varepsilon\left|\mathbf{H}(\delta\mathbf{x})\right|}=1+O(\varepsilon)\,, \tag{30}$$

where $H$ and $\mathbf{H}$ represent the nonlinear operator and the corresponding TL version, respectively, $\mathbf{x}$ is the vector of the model state variables ($q_r$, $q_s$, $q_g$), whose perturbation is denoted by $\delta\mathbf{x}$, and $\varepsilon$ is the perturbation magnitude and is greater than zero; $\delta\mathbf{x}$ used the perturbations generated for all_prt. Table 3 shows that the tangent linear operator is sufficiently accurate with a ratio close to 1.0.

A correct adjoint operator must satisfy the relationship that is given by

$$(\mathbf{H}\delta\mathbf{x})^{\mathrm{T}}\mathbf{H}\delta\mathbf{x}=\delta\mathbf{x}^{\mathrm{T}}\mathbf{H}^{\mathrm{T}}(\mathbf{H}\delta\mathbf{x})\,. \tag{31}$$

The meanings of all of the symbols in this equation are consistent with those in Eq. (30), and $\mathbf{H}^{\mathrm{T}}$ is the adjoint operator. All of the perturbations used for the tangent linear test were adopted in the adjoint test. In the double precision test, there were 14 identical digits on both sides of Eq. (31); in the single precision test, there were 7 identical digits.

## 5 Results

### 5.1 Analyses in the observation space and the model space

The Exp_ref analyses (6 outer loops, 150 inner iterations, and 0.1 weight for the retrieval part) are first examined in both the reflectivity space and model space. As expected, the analyzed reflectivity agrees much more closely with the observed reflectivity than the background reflectivity for the both analysis times (00Z and 01Z) (Fig. 6). In addition, considering that the 00Z background bias is much larger than that at 01Z, the comparable analysis error for both times indicates that the

analysis is generally relatively insensitive to the initial background. However, a small number of points in the analysis have zero reflectivity, while the corresponding observations can be greater than 10 dBZ (Fig. 6b, d). Further examination indicates that these failed points are related to the locations with very small background values of $q_r$, $q_s$, and $q_g$, where the nonlinearity of the radar operator and the deficiency of the TL/AD operator are more pronounced.

Figure 7 shows the horizontal distributions of the observed and analyzed reflectivities at 2 km, 4 km (melting layer), and 6

km AGL. In general, they match well in observed areas with a weaker analysis at 00Z, which is likely caused by the relatively bad background. Spurious echoes appear over unobserved areas in the analyses at both 00Z and 01Z and most likely resulted from the spatial correlations in the background error covariance; these correlations allow the propagation of information from observed areas to unobserved areas both horizontally and vertically. This result implies that the statistical correlation length scales obtained from a one-month forecast sample may be too large for this squall line case. The results of

tuning the horizontal correlation length scales will be given in Section 5.4.



The 00Z and 01Z analyses of $q_r$, $q_s$, and $q_g$ at three model levels are shown in Fig. 8. The direct assimilation of the reflectivity data using RadZIceVar successfully retrieved the lower level rain, the upper level snow or graupel, and mixed rain/snow and/or rain/graupel in the melting layer (model level 15). Note that the analysis increment can be created only at levels where the STD of the background error is larger than zero (see Fig. 3).

## 5.2 Convergence of the minimization

Figure 9 shows the cost function and the norm of its gradient as a function of the number of inner iterations for the 1st, 3rd, and 6th outer loops. For the 00Z Exp_ref analysis, both the cost function and the gradient norm decrease rapidly in the first outer loop due to the large adjustment from the bad initial background. Using the improved guess after 2 outer loops, the third outer loop starts from an ~85% smaller cost function, which is then reduced more gradually with increasing iterations. Performing more outer loops does not further reduce the cost function substantially, and outer loop 6 even has a slightly larger cost function than that of outer loop 3. Similar behavior can also be observed in the 01Z analysis, but performing more than three outer loops can further reduce the cost function to a small extent. This indicates that performing six outer loops may not be necessary and that 2 or 3 outer loops may be sufficient.

To more quantitatively measure the impacts of the number of outer loops and inner iterations, the correlation coefficients between the Exp_ref analysis (i.e., 6 outer loops and 150 iterations in each loop) and the analyses with 1-5 outer loops and 50/100/150 inner iterations in each loop are calculated and shown in Fig. 10. In the 00Z analysis, the correlation coefficients of $q_r$ and $q_s$ are already greater than 0.8 after the first outer loop, and using 50 or 100 inner iterations also results in good agreement with the Exp_ref analysis for $q_r$ and $q_s$. However, this is not the case for $q_g$, which is likely an indication of a bad $q_g$ background (0.1 of the retrieved values). With a more realistic background at 01Z, the issue of the $q_g$ analysis is not pronounced (Fig. 10b), and the analysis with three outer loops and 100 inner iterations in each loop gives coefficients greater than 0.8 for both $q_r$ and $q_s$, further confirming the importance of the background quality.

## 5.3 Importance of hydrometeor preprocessing

To further investigate the sensitivity of the analysis to the background quality, additional analyses without the hydrometeor preprocessing step were performed at 00Z and 01Z. At 00Z, the nonzero background was constructed by using a very small weight ($5\times10^{-4}$) for the retrievals. At 01Z, the background is simply taken from the 1-h WRF forecast. Compared to Fig. 6b, the analysis at 00Z (Fig. 11b) has many more failed points (i.e., those with zero analyzed reflectivity) due to the excessively small background hydrometeor values. In contrast, the 01Z analysis (Fig. 11d) without preprocessing has a comparable error bias and STD to those with the preprocessing (Fig. 6d). These results indicate that the preprocessing step is mostly needed to address the bad background.

Fig. 12a shows that the analyzed reflectivity in the melting layer has a reasonable fit to the observations even though it started from very small values of the background reflectivity. However, the analysis in the model space (Fig. 12b) is very different from that with the preprocessing (Fig. 8b). With the preprocessing step, the analyzed melting layer is mostly a




mixture of rainwater and snow, while a three-phase mixture of hydrometeors is present when no preprocessing is applied. Note that the hydrometeor's phase information could be better determined by assimilating polarization measurements from dual-polarization radar, which will be explored in a future study.

In contrast, the 01Z analysis without the preprocessing step (Fig. 12c, d) closely resembles that with the preprocessing (Fig. 7h and Fig. 8e) in both the observation and model spaces, especially over the strong convective line (black dashed line in Fig. 12d). However, removing this preprocessing step results in a spurious strong echo (marked by "A" in Fig. 12c) that is much weaker in Fig. 7h. This implies that the preprocessing step will still be helpful for some "bad" locations (e.g., mismatched convective cells between the model background and observation) for a generally "good" background.

## 5.4 Impact of the spatial correlation scale

Figure 13 shows the 00Z analysis reflectivity at 4 km AGL with the horizontal correlation length scales of the background errors reduced by factors of 2 and 4. The spurious echoes weaken as the length scale decreases, especially for the strong spurious echo marked by "A". In addition, reducing the length scale improves the intensity analysis at certain locations (e.g., the convective cell marked by "B"; also see Fig. 7e, 7f). Note that the background error statistics used for these 3DVAR analyses were obtained from forecast samples over a month-long period and are likely not optimal for this particular squall line case. A better solution would be the use of the flow-dependent background error covariance, which will be investigated in the future using more advanced DA methods, such as 4DVAR and hybrid-3D/4DEnVar that are available in WRFDA.

## 5.5 Impact on the precipitation forecast

The performances of the precipitation forecasts were quantitatively evaluated using the Stage IV dataset with the fractions skill score (FSS; Roberts and Lean, 2008). Hourly precipitation forecasts between 02Z and 05Z were evaluated because assimilating only the reflectivity is expected to have an impact mostly on the short-term forecast. Similar to Schwartz et al. (2014), the aggregated FSS over the period from 02Z to 05Z is shown in Fig. 14 for the forecasts initialized at 00Z and 01Z and for different rainfall thresholds. For the forecast from 00Z (Fig. 15a), Exp_ref obtained greater FSSs than those of noDA for all thresholds with a radius of influence smaller than 50 km. With a larger radius, this difference remains for the light rain ($\leq$ 5 mm) but disappears for the heavier rain. Similar behavior can be observed for the forecast at 01Z (Fig. 15b) but with a more positive impact from the radar DA for heavier rainfall and for a radius of influence greater than 50 km. Further examination (not shown) indicated that the improvement in the light rain prediction was associated with the larger snow area in the analysis, so the light rain missed by noDA was better captured in Exp_ref. This examination also showed that the higher FSSs obtained by Exp_ref for the heavier rainfall were mostly associated with the smaller displacement error.





## 6 Conclusions

This study developed tangent linear and adjoint operators based on the J08 reflectivity operator and implemented them in WRFDA. This new operator can compute the reflectivity contributed by ice-phase species and is called RadZIceVar. RadZIceVar is effective at analyzing rainwater, snow, and graupel by directly assimilating US WSR-88D S-band radar

reflectivity with WRFDA's 3DVAR. The analysis accuracy is somewhat sensitive to the numbers of outer loops and inner iterations. The results indicated that 2-3 outer loops with 50-100 iterations in each loop are needed to obtain a sufficiently accurate analysis. Two deficiencies are observed in the 3DVAR analysis. One issue is the analysis failures at locations with observed radar echoes but with zero or excessively small model background values of the hydrometeors. This issue can be partially resolved using a preprocessing step with radar-retrieved hydrometeors to improve the "bad" background before the

analysis. Another issue is the spurious radar echoes (i.e., precipitation) in the analysis caused by the spatial correlations in the background error covariance. These can be reduced by decreasing the correlation length scales. In addition, the short-term (2 h – 5 h) precipitation forecast is improved by the direct reflectivity DA even though the inexpensive univariate 3DVAR technique is used in this first attempt of applying RadZIceVar.

A more thorough evaluation of reflectivity DA with RadZIceVar will be examined in a future study using a more advanced

hybrid ensemble-variational DA technique, which allows flow-dependent background errors with multivariate correlations and is expected to further reduce the aforementioned deficiencies. Moreover, RadZIceVar will be extended to include the computation of polarimetric quantities to better determine the phases of the hydrometeors, especially in the melting layers.

## Code and data availability

The RadZIceVar v1.0 operator is integrated into the community WRFDA software and will be publicly available in a future

release. The data used in this study can be obtained by contacting the corresponding author via e-mail.

## Author contributions

Shizhang Wang performed the coding and designed the data assimilation experiments. Zhiquan Liu supervised this study. All authors contributed to the writing of the paper.

## Acknowledgements

This work was jointly sponsored by the National Key Research and Development Program of China (2017YFC1502103), the National Natural Science Foundation of China (41505089, 41875129, 41505090, 41430427, and 41805070), the Startup



Foundation for Introducing Talent of NUIST (2014R007), and National Key Research and Development Program of China (2018YFC1506404). NCAR is sponsored by the US National Science Foundation.

**Appendix**

This section provides details about the reorganization of all of the terms in the brackets in Eq. (11) in terms of the $f_{wx}$ power.

For convenience, the expressions in these brackets are represented by G($f_{wx}$).

Applying Eq. (16) to the brackets in Eq. (11) results in

$$G(f_{wx}) = \varepsilon_x^2 [A(\sum_{k=0}^{n} P_{wxak} f_{wx}^k)^2 + B(\sum_{k=0}^{n} P_{wxbk} f_{wx}^k)^2 + 2C(\sum_{k=0}^{n} P_{wxak} f_{wx}^k)(\sum_{k=0}^{n} P_{wxbk} f_{wx}^k)] \cdot \qquad \text{A. 1}$$

Expanding all of the terms of A. (1) results in

$$G(f_{wx}) = \varepsilon_x^2 \{ A\sum_{i=0}^{n} [P_{wxai}(\sum_{j=0}^{n} P_{wxaj} f_{wx}^{i+j})] + B\sum_{i=0}^{n} [P_{wxbi}(\sum_{j=0}^{n} P_{wxbj} f_{wx}^{i+j})] + 2C\sum_{i=0}^{n} [P_{wxai}(\sum_{j=0}^{n} P_{wxbj} f_{wx}^{i+j})] \} \cdot \qquad \text{A. 2}$$

Reorganizing the third term (omitting 2C) of A. (2) in terms of the $f_{wx}$ power ($i+j$) results in

$$
\begin{aligned}
\sum_{i=0}^{n} [P_{wxai}(\sum_{j=0}^{n} P_{wxbj} f_{wx}^{i+j})] =\ & P_{wxa0} P_{wxb0} f_{wx}^0 \\
& + P_{wxa0} P_{wxb1} f_{wx}^1 + P_{wxa1} P_{wxb0} f_{wx}^1 \\
& \cdots \\
& + P_{wxa0} P_{wxbn} f_{wx}^n + P_{wxa1} P_{wxb(n-1)} f_{wx}^n + \cdots + P_{wxan} P_{wxb0} f_{wx}^n \\
& + P_{wxa1} P_{wxbn} f_{wx}^{n+1} + P_{wxa2} P_{wxb(n-1)} f_{wx}^{n+1} + \cdots + P_{wxan} P_{wxb1} f_{wx}^{n+1} \\
& \cdots \\
& + P_{wxa(n-1)} P_{wxbn} f_{wx}^{2n-1} + P_{wxan} P_{wxb(n-1)} f_{wx}^{2n-1} \\
& + P_{wxan} P_{wxbn} f_{wx}^{2n} \\
=\ & \sum_{k=0}^{n} [f_{wx}^k \sum_{i=0}^{k} P_{wxai} P_{wxb(k-i)}] + \sum_{k=n+1}^{2n} [f_{wx}^k \sum_{i=k-n}^{n} P_{wxai} P_{wxb(k-i)}]
\end{aligned}
\qquad \text{A. 3}
$$

The sum functions in the square brackets on the right-hand side of A. (3) correspond to the third expression of Eq. (20). Using the third expression of Eq. (20), the third term of A. (2) is rewritten as follows,

$$
\begin{aligned}
2C\sum_{i=0}^{n} [P_{wxai}(\sum_{j=0}^{n} P_{wxbj} f_{wx}^{i+j})] =\ & 2C\{ \sum_{k=0}^{n} [f_{wx}^k \sum_{i=0}^{k} P_{wxai} P_{wxb(k-i)}] + \sum_{k=n+1}^{2n} [f_{wx}^k \sum_{i=k-n}^{n} P_{wxai} P_{wxb(k-i)}] \} \\
=\ & 2C\sum_{k=0}^{2n} P_{Cxk} f_{wx}^k
\end{aligned}
\qquad \text{A. 4}
$$

where $P_{Cxk}$ has the same meaning as in Eq. (20). Similarly, we can rewrite the other two terms in A. (2) as follows:





$$A\sum_{i=0}^{n}[P_{\text{wxa}i}(\sum_{j=0}^{n}P_{\text{wxa}j}f_{\text{wx}}^{i+j})]=A\{\sum_{k=0}^{n}[f_{\text{wx}}^{k}\sum_{i=0}^{k}P_{\text{wxa}i}P_{\text{wxa}(k-i)}]+\sum_{k=n+1}^{2n}[f_{\text{wx}}^{k}\sum_{i=k-n}^{n}P_{\text{wxa}i}P_{\text{wxa}(k-i)}]\}=A\sum_{k=0}^{2n}P_{\text{Ax}k}f_{\text{wx}}^{k} \quad . \qquad \text{A. 5}$$

$$B\sum_{i=0}^{n}[P_{\text{wxb}i}(\sum_{j=0}^{n}P_{\text{wxb}j}f_{\text{wx}}^{i+j})]=B\{\sum_{k=0}^{n}[f_{\text{wx}}^{k}\sum_{i=0}^{k}P_{\text{wxb}i}P_{\text{wxb}(k-i)}]+\sum_{k=n+1}^{2n}[f_{\text{wx}}^{k}\sum_{i=k-n}^{n}P_{\text{wxb}i}P_{\text{wxb}(k-i)}]\}=B\sum_{k=0}^{2n}P_{\text{Bx}k}f_{\text{wx}}^{k}$$

Because these three expressions (A. (4) and A. (5)) contain the same sum function with respect to $k$ from 0 to $2n$, A. (2) can be rewritten as follows:

$$
\begin{aligned}
G(f_{\text{wx}}) &= \varepsilon_{\text{x}}^{2}(A\sum_{k=0}^{2n}P_{\text{Ax}k}f_{\text{wx}}^{k}+B\sum_{k=0}^{2n}P_{\text{Bx}k}f_{\text{wx}}^{k}+2C\sum_{k=0}^{2n}P_{\text{Cx}k}f_{\text{wx}}^{k}) \\
&= \varepsilon_{\text{x}}^{2}\sum_{k=0}^{2n}(AP_{\text{Ax}k}f_{\text{wx}}^{k}+BP_{\text{Bx}k}f_{\text{wx}}^{k}+2CP_{\text{Cx}k}f_{\text{wx}}^{k}) \qquad . \qquad \text{A. 6}\\
&= \varepsilon_{\text{x}}^{2}\sum_{k=0}^{2n}(AP_{\text{Ax}k}+BP_{\text{Bx}k}+2CP_{\text{Cx}k})f_{\text{wx}}^{k} \\
&= \varepsilon_{\text{x}}^{2}\sum_{k=0}^{2n}P_{\text{x}k}f_{\text{wx}}^{k}
\end{aligned}
$$

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

**Tables**

**Table 1 Values of $\alpha_{dsa}$, $\alpha_{dsb}$, $\alpha_{dga}$, and $\alpha_{dgb}$.**

| $\alpha_{dsa}$ | $\alpha_{dsb}$ | $\alpha_{dga}$ | $\alpha_{dgb}$ |
|---|---|---|---|
| $0.194 \times 10^{-4}$ | $0.191 \times 10^{-4}$ | $0.191 \times 10^{-3}$ | $0.165 \times 10^{-3}$ |



**Table 2 Values of $P_{wxak}$ and $P_{wxbk}$ in Eq. (16).**

| $k$ | 0 | 1 | 2 | 3 | 4 | 5 | 6 |
|---|---|---|---|---|---|---|---|
| $P_{wsak}$ | 0.194 | 7.094 | 2.135 | -5.225 | 0 | 0 | 0 |
| $P_{wsbk}$ | 0.191 | 6.916 | -2.841 | -1.160 | 0 | 0 | 0 |
| $P_{wgak}$ | 0.191 | 2.39 | -12.57 | 38.71 | -65.53 | 56.16 | -18.98 |
| $P_{wgbk}$ | 0.165 | 1.72 | -9.92 | 32.15 | -56.0 | 48.84 | -16.69 |

**Table 3 Perturbation samples for the verification of the tangent linear operator.**

|  | $q_r$ (kg kg$^{-1}$) | $q_s$ (kg kg$^{-1}$) | $q_g$ (kg kg$^{-1}$) | ratio |
|---|---|---|---|---|
| sample 1 | min: $-4.1 \times 10^{-4}$ <br> max: $4.7 \times 10^{-4}$ | min: $-4.2 \times 10^{-4}$ <br> max: $4.4 \times 10^{-4}$ | min: $-3.7 \times 10^{-4}$ <br> max: $3.1 \times 10^{-4}$ | 1.00114799 |
| sample 2 | min: $-1.6 \times 10^{-6}$ <br> max: $1.8 \times 10^{-6}$ | min: $-1.6 \times 10^{-6}$ <br> max: $1.7 \times 10^{-6}$ | min: $-1.4 \times 10^{-6}$ <br> max: $1.2 \times 10^{-6}$ | 1.00004709 |





**Figures**

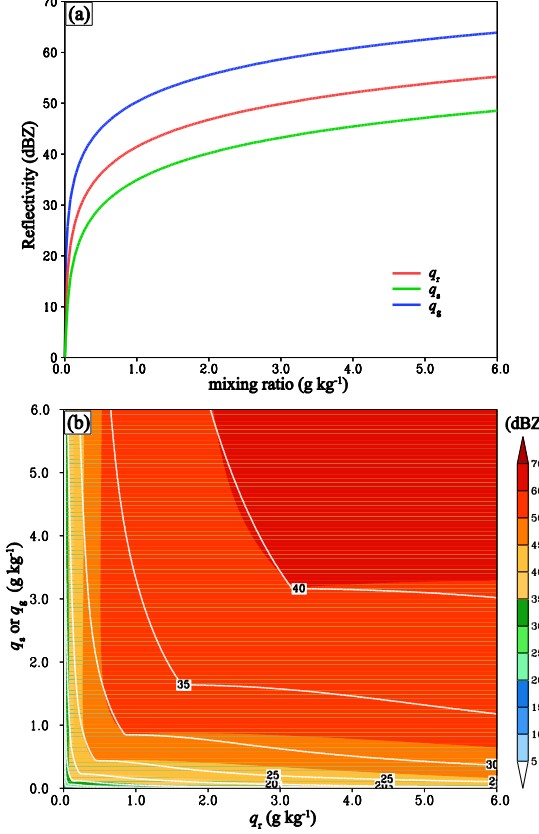

**Fig. 1 (a) The reflectivity as (a) a function of $q_r$ (red), $q_s$ (green), and $q_g$ (blue) for pure water and dry snow/graupel and (b) a function of $q_r$–$q_s$ (colors) and $q_r$–$q_g$ (contours) for wet snow/graupel.**




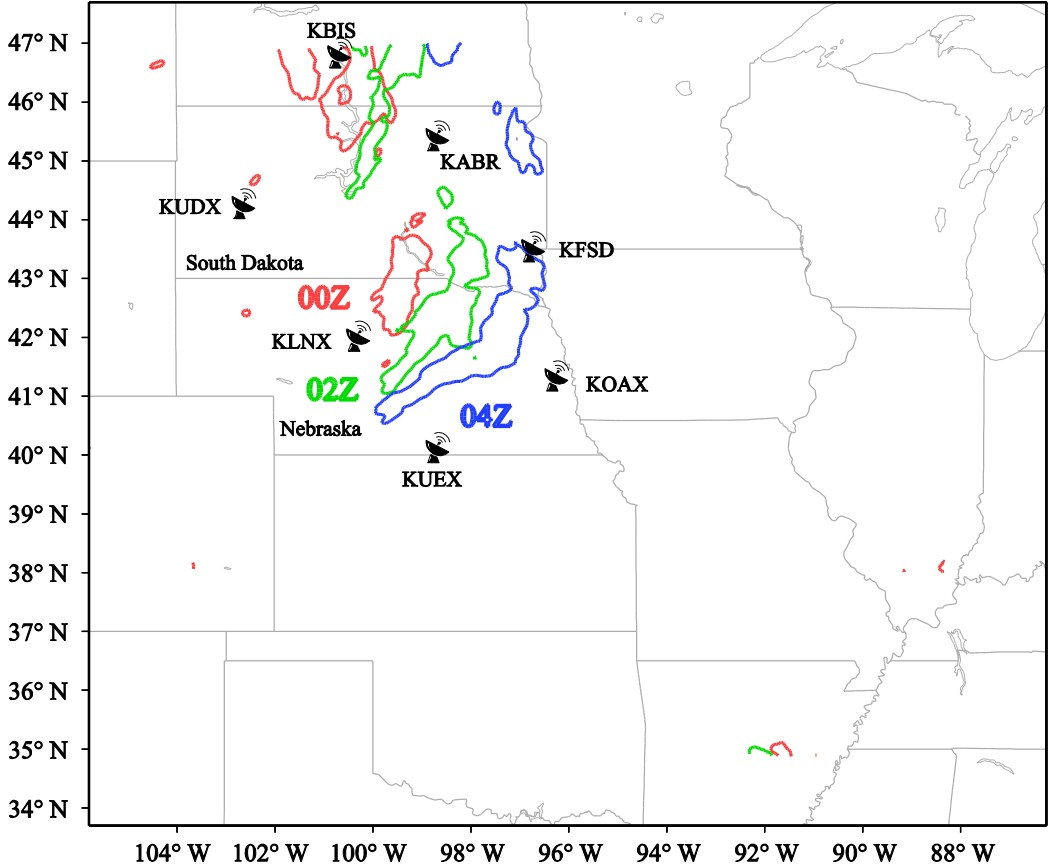

**Fig. 2 The simulation domain with radar sites marked by radar icons and names. The areas of precipitation greater than 5 mm h⁻¹ are plotted for 00Z (red), 02Z (green), and 04Z (blue) on 2 June 2018.**





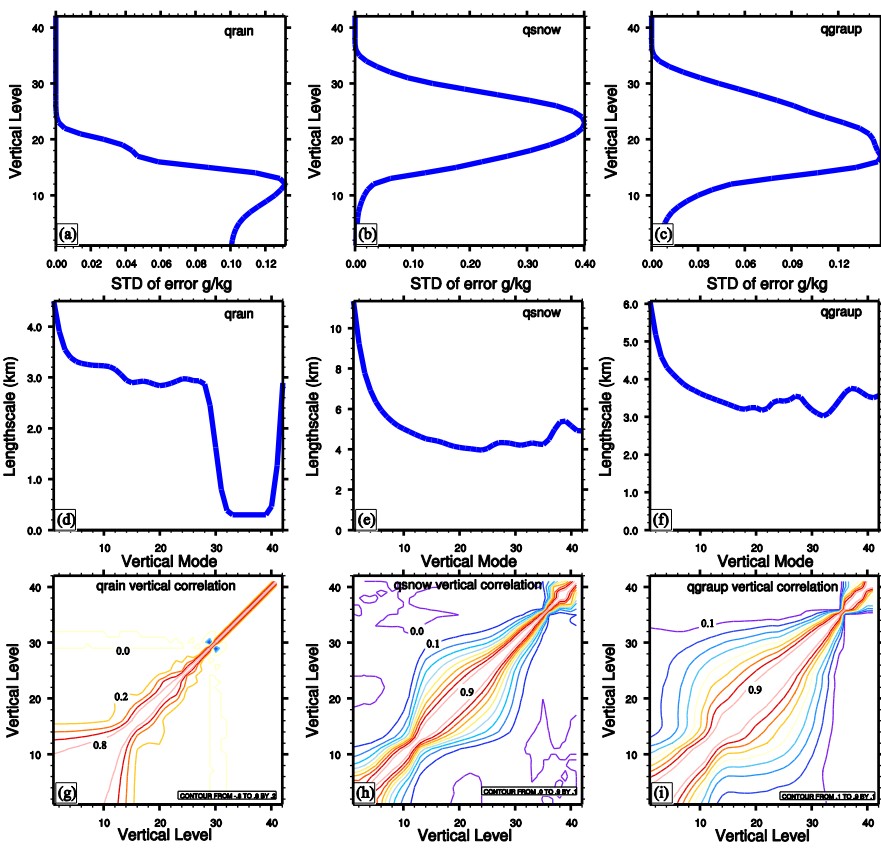

**Fig. 3 (a-c) The standard deviations of the background errors at different vertical levels, (d-f) the horizontal correlation length scale as a function of EOF mode, and (g-i) the vertical correlation coefficients for $q_r$, $q_s$, and $q_g$.**



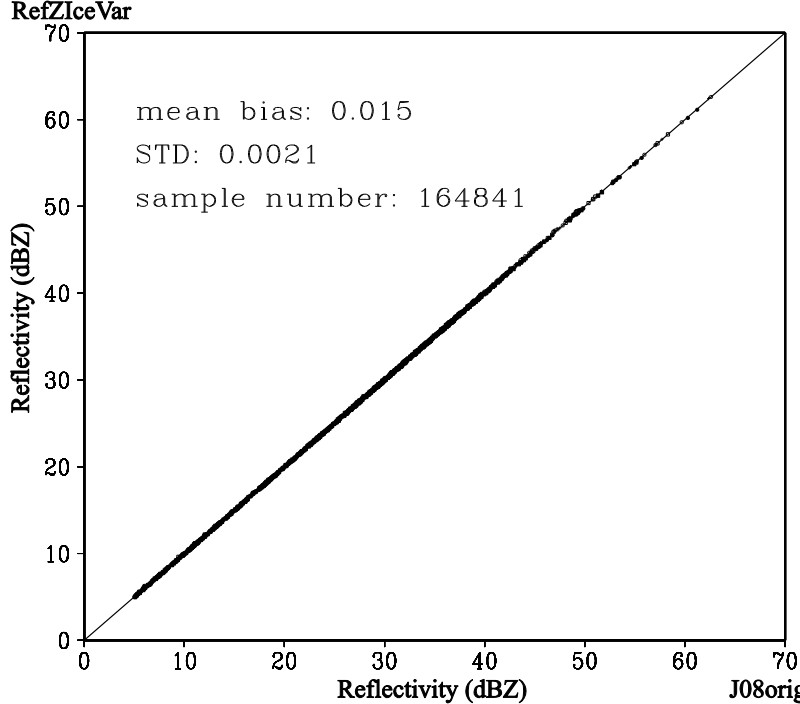

**Fig. 4 Scatter plot of the reflectivity for J08orig (*x* axis) versus RadZIceVar (*y* axis). The bias, standard deviation (STD), and number of samples are listed in the plot.**





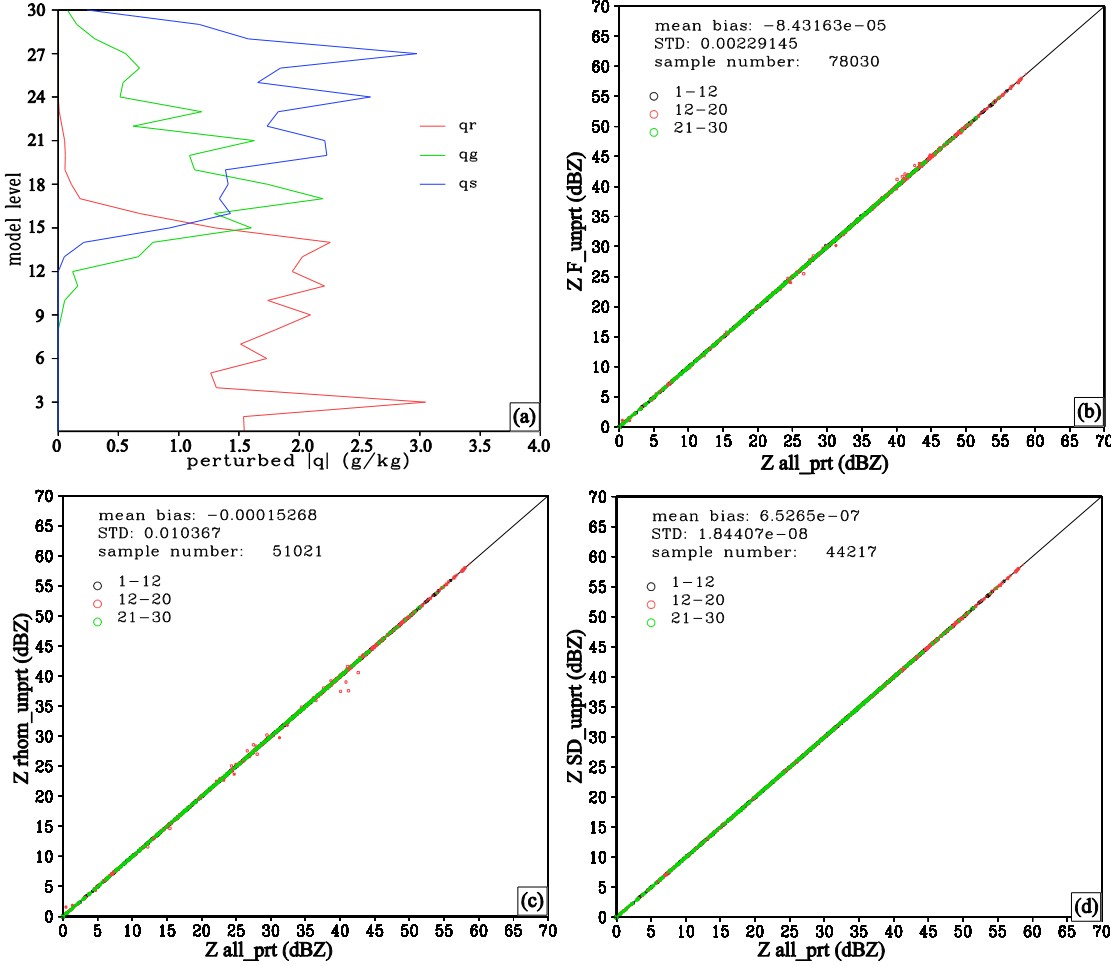

**Fig. 5 (a) Vertical distributions of the maximum absolute values of the perturbed** $q_r$ **(red),** $q_s$ **(green), and** $q_g$ **(blue). Reflectivity scatter plots of all_prt (*x* axis) versus (b) F_unprt (*y* axis), (c) rhom_unprt (*y* axis), and (d) SD_unprt (*y* axis) at model levels 1~12 (black, lower than 3 km AGL), 12~20 (red, between 3~7 km AGL), and 21~30 (green, above 7 km AGL).**





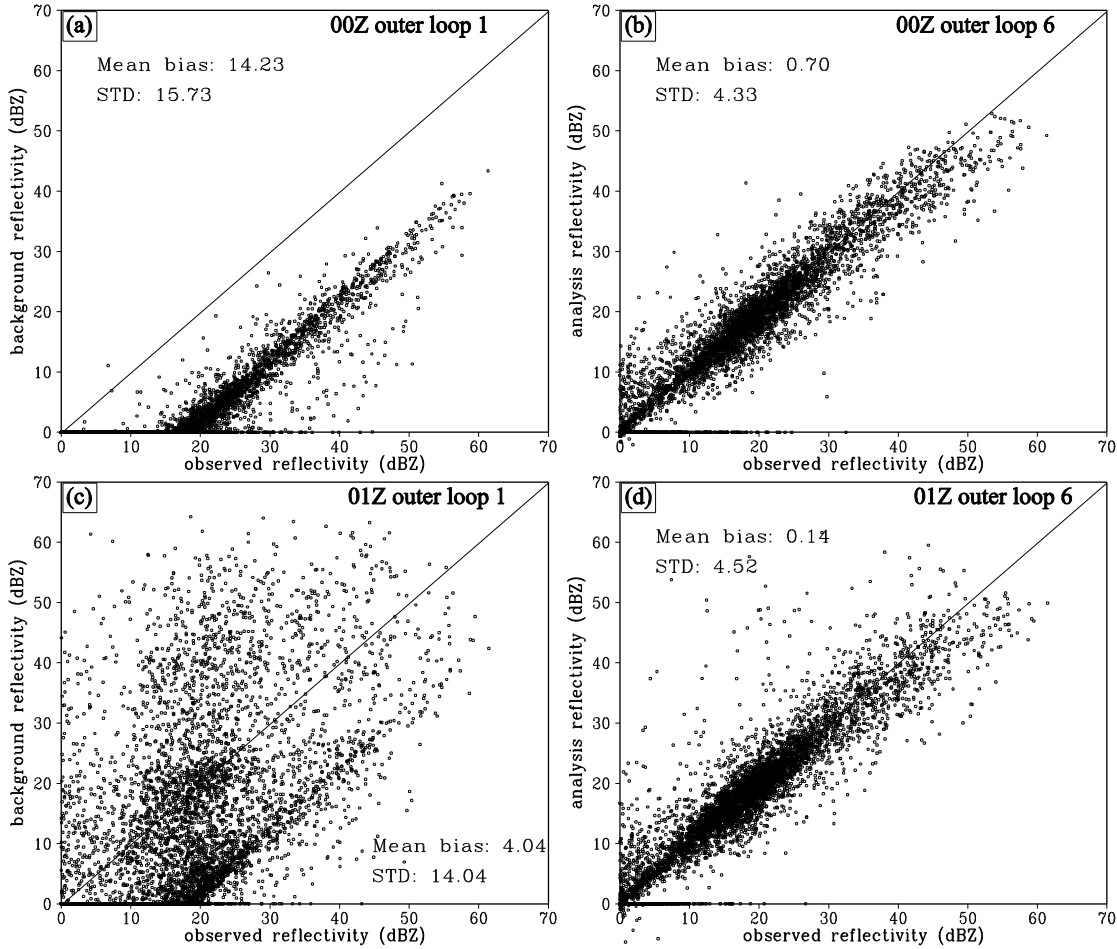

**Fig. 6 Scatter plots of the observed (*x* axis) versus (a, c) background and (b, d) analysis reflectivity at (a, b) 00Z and (c, d) 01Z in Exp_ref. The mean bias and standard deviation (STD) between the observations and the background (or analysis) are listed in each plot.**



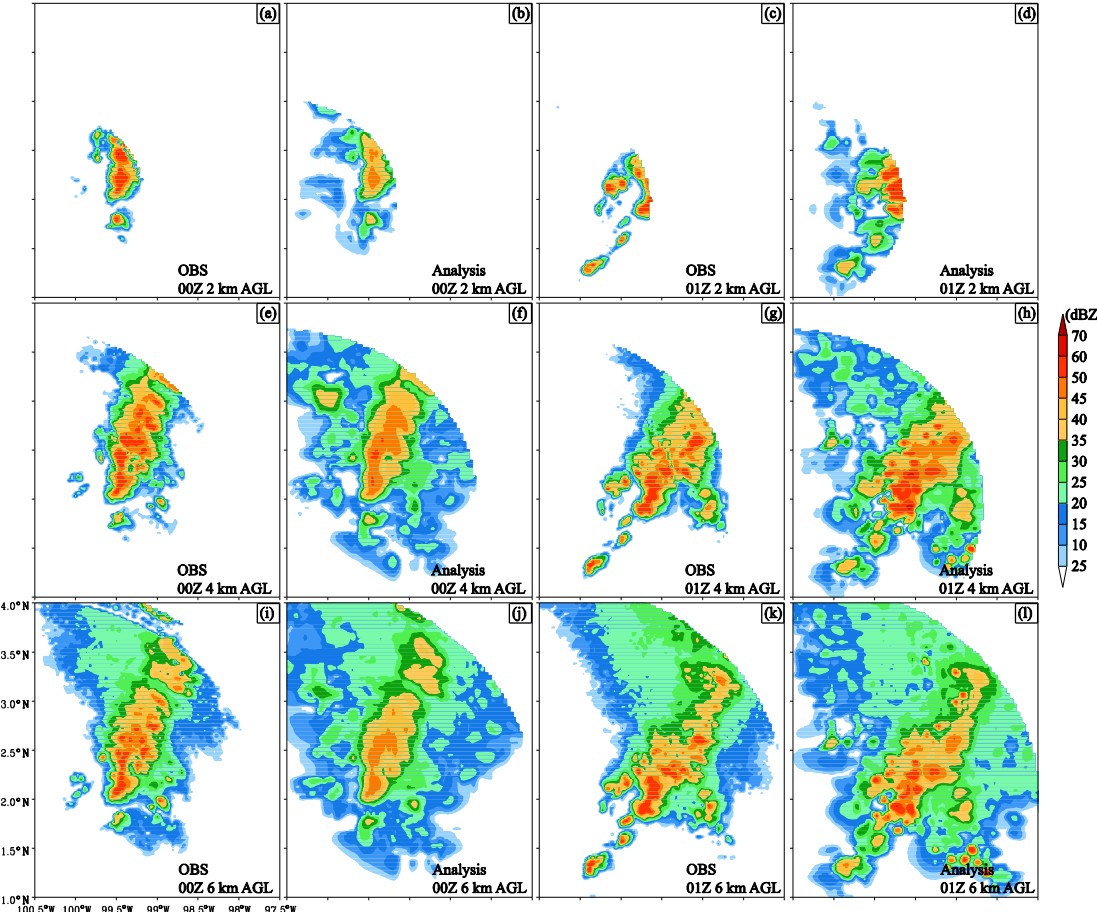

**Fig. 7** The (a, e, i, c, g, k) observed (KLNX) and (b, f, j, d, h, l) analysis reflectivities at (a-d) 2 km, (e-h) 4 km, and (i-l) 6 km AGL at (a, e, i, b, f, j) 00Z and (c, g, k, d, h, l) 01Z in Exp_ref.





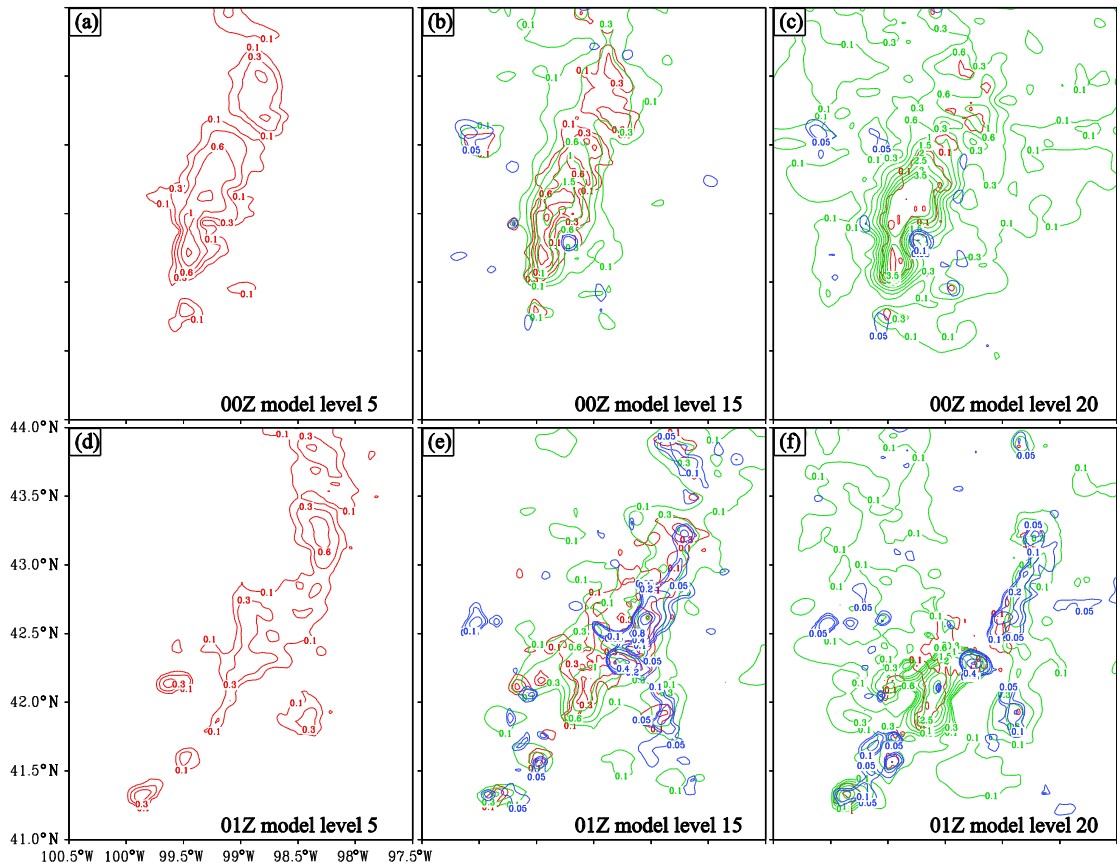

**Fig. 8 The (a, b, c) 00Z and (d, e, f) 01Z analyses of $q_r$ (red), $q_s$ (green), and $q_g$ (blue) at model levels 5 (left column), 15 (middle column), and 20 (right column) for outer loop 6 in Exp_ref. Model levels 5, 15, and 20 approximately correspond to 0.7 km, 4.0 km, and 8.0 km AGL, respectively.**





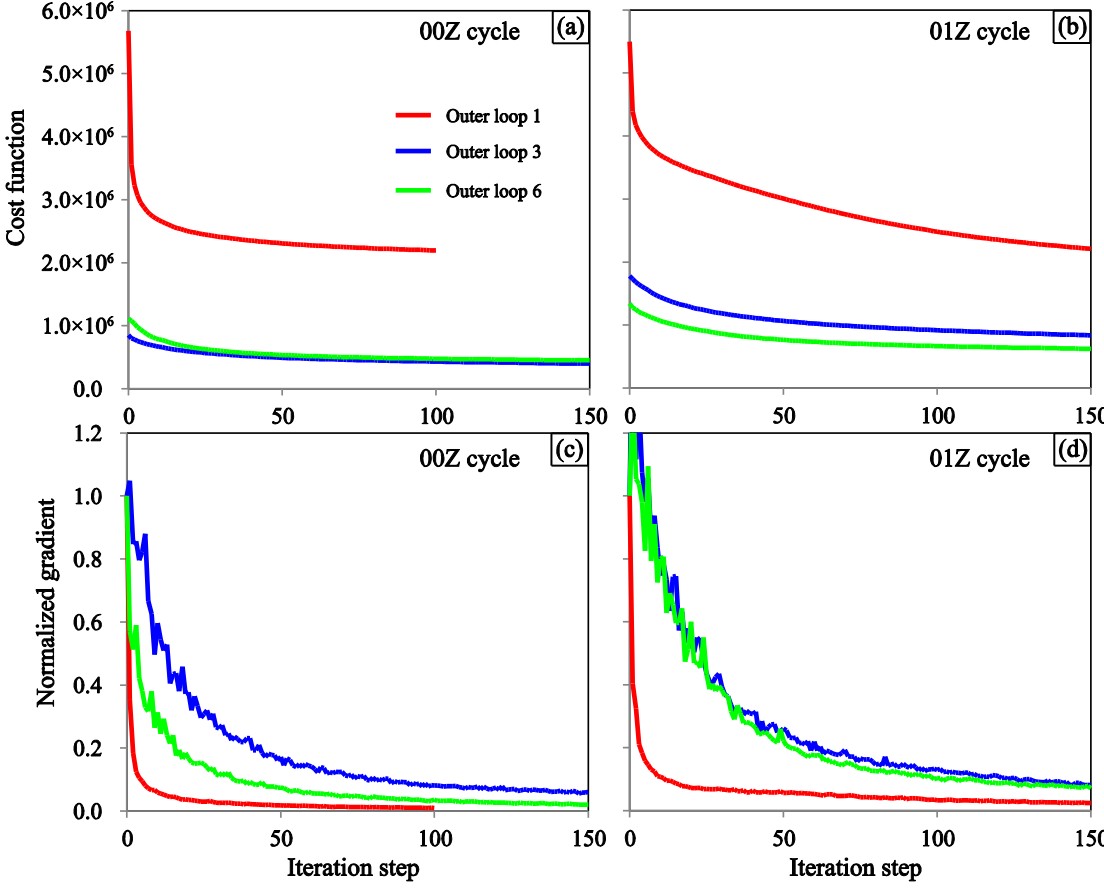

Fig. 9 The cost function and normalized gradient norm as functions of the iteration step during the minimization at (a, c) 00Z and (b, d) 01Z in Exp_ref.





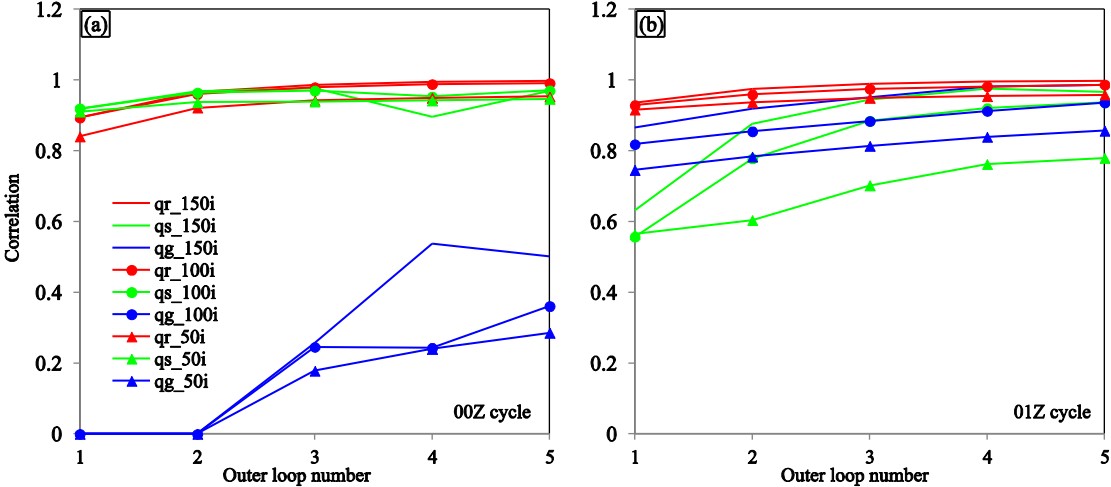

**Fig. 10 The correlation coefficients of $q_r$ (red), $q_s$ (green), and $q_g$ (blue) between the Exp_ref analysis (with six outer loops and 150 inner iterations each loop) and that with different numbers of outer loops and inner iterations at (a) 00Z and (b) 01Z.**



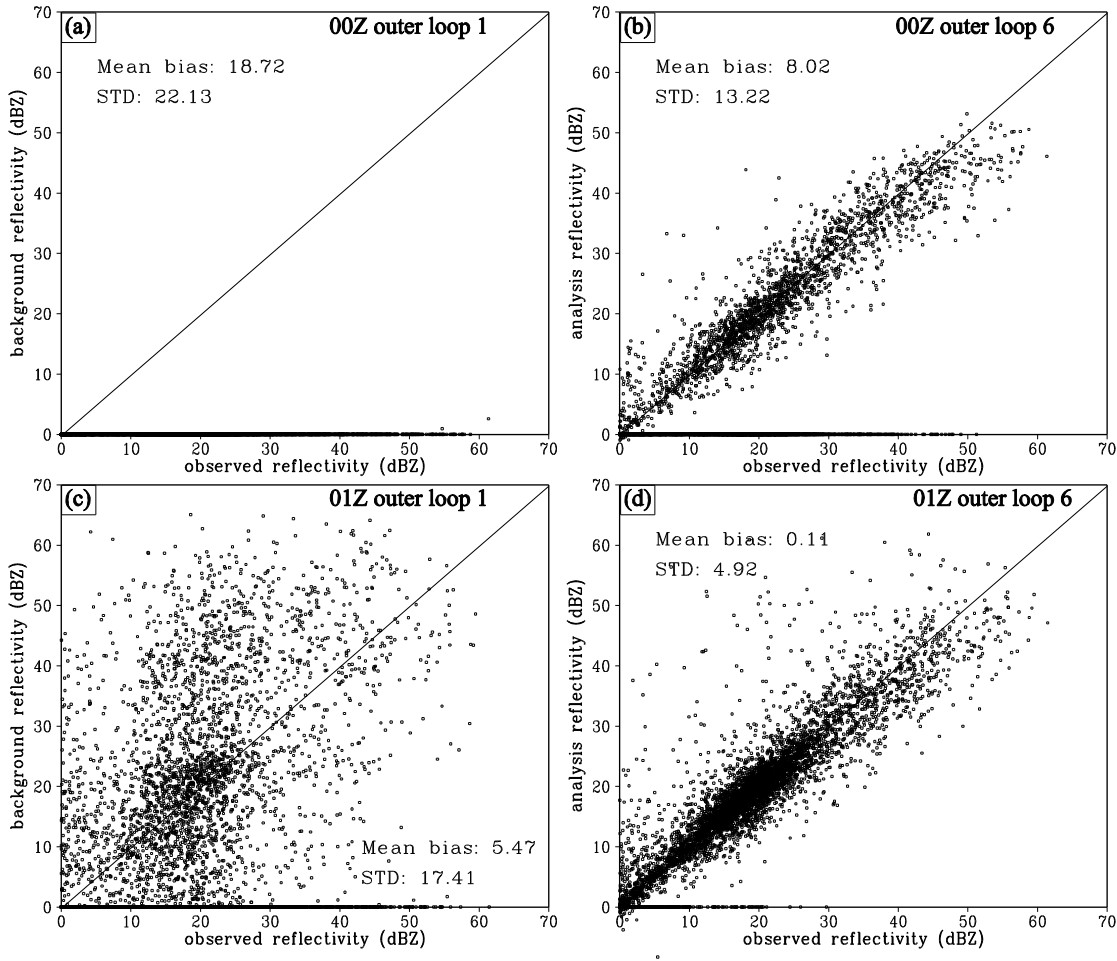

**Fig. 11 Same as Fig. 6 but for the experiments without hydrometeor preprocessing at (a, b) 00Z and (c, d) 01Z.**



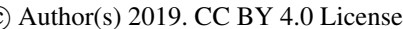

**Fig. 12 (a, c) The analysis reflectivity at 4 km AGL and (b, d) the analyses of $q_r$ (red), $q_s$ (green), and $q_g$ (blue) at model level 15 at 00Z and 01Z for the experiments without hydrometeor preprocessing.**





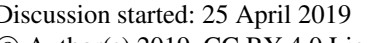

**Fig. 13 Same as Fig. 7 but for (a) the observations and the analyses from (b) Exp_ref, (c) Exp_ls0.5, and (d) Exp_ls0.125.**





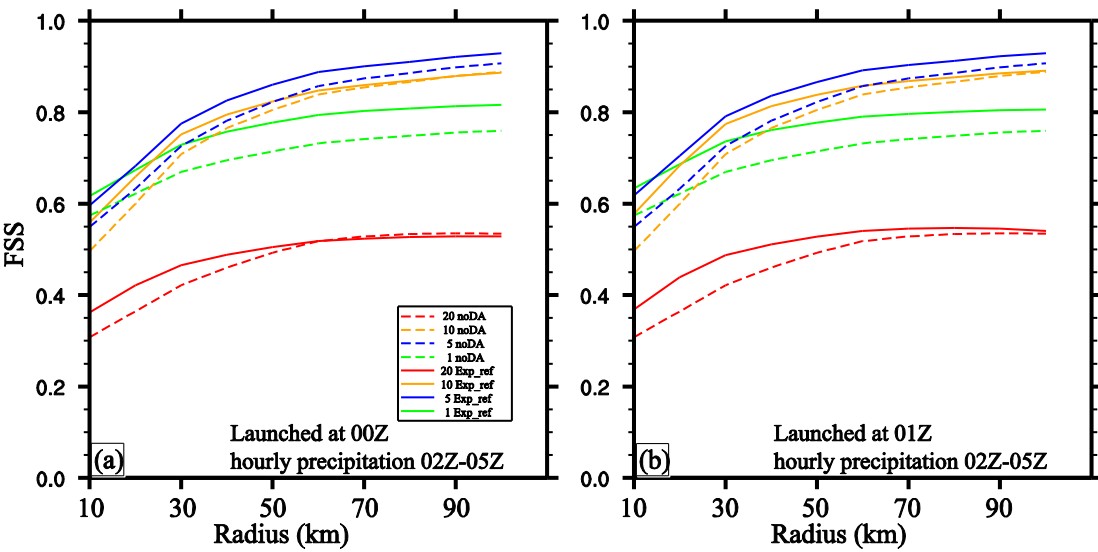

**Fig. 14 The FSSs of the hourly precipitation forecasts aggregated over the period from 02Z to 05Z as functions of the radius of influence for forecasts launched from the (a) 00Z and (b) 01Z analyses for Exp_ref (solid lines) and noDA (dashed lines). The hourly precipitation thresholds are denoted by green (1 mm), blue (5 mm), orange (10 mm), and red (20 mm) lines.**

