# Peer review of "A Radar Reflectivity Operator with Ice-Phase Hydrometeors for Variational Data Assimilation (RadZIceVar v1.0) and Its Evaluation with Real Radar Data"

_Geoscientific Model Development, 2019_

## Referee Comment (RC1) · Juanzhen Sun (Referee) · 22 May 2019

This paper describes an radar reflectivity operator and its adjoint for variational data assimilation. The new operator is implemented in WRFDA and preliminary test is presented with a convective case occurred in the U.S.. The developmental procedures of the operator is well described and the results are well explained and analyzed. My recommendation is to publish on GMD after minor revision. Although I do not have any major concerns about the content of the paper, I do suggest that the authors pay good attention to improve the English writing. Below are some suggestions from me to

help improve the readability of the paper, but I strongly suggest the authors to hire a professional editor to further improve the paper.

1. Title: The abbreviated algorithm name is not necessary in the title. 2. Abstract, lines 15-18: Change to "It is shown that the deficiencies in the analysis using this operator, caused by the poor quality….error covariance, can be partially resolved ….". 3. Page 1, lines 24-25: "…Xue et al., 2006) and they have demonstrated that assimilating these observations improves…" 4. Page 2, line 1: Change "limited in" to "limited to". 5. Page 2, line 12: Add references for this statement. 6. Page 2, line 13: Change "and the conditions that" to "in which". 7. Page 2, line 22: Change this sentence to "Reflectivity operators have been developed both for the variational method (….) and for the ensemble Kalman filter method…". 8. Page 2, line 30: Suggested change: "Despite the difficulty, some efforts have been undertaken for reflectivity assimilation ….." 9. Page 3, line 14: "To compute Eq. (2), the mixing ratios of ….. are required." 10. Sections 2.1.2 and 2.1.3: There are so many parameters, such as those in Eqs. (11) and (16)-(20). Can you briefly explain the meanings of these parameters? Are they theoretically or empirically determined? What are their uncertainties? 11. Page 9, line 14: "…more substantially than that from dry snow…" 12. Page 9, line 24: "….from southern South Dakota to northern Nebraska, as shown in Fig.2. Note that there is also a weaker precipitation system near the north boundary of the domain. The top of the convective system of interest at this time, identified by reflectivity greater than 5 dBZ, reached 16 AGL". In the next line, "a bow echo was observed…". Line 3: delete "as shown in Fig. 2" here. 13. Page 10, line 25: Change "that of snow and graupel" to "those of snow and graupel". Line 27: do you mean a broad vertical distribution? 14. Page 11, line 25: "More outer loops were necessary due to the inaccurate …". Lines 26-28: do you mean a total of four experiments were performed by varying the number of iterations and the analysis time? 15. Page 11, line 30: "RadZIceVar is unable to create hydrometeor increments…"? 16. Page 12, line 2: "…constant, it is expected that…". This should be a general expectation, so you do not have to refer to J08. Line 7: "nonzero" instead of "nonvanishing". Line 9: "To examine the analysis

performance...". Line 14: add "the" before "length scale". 17. Page 13, line 16: Why was such a small weighting 0.1 used? Did you tried any other weighting? 18. Page 13, line 17: The use of "reflectivity space" and "model space" are not appropriate in this context. Also in line 5 on page 15. Line 18: delete "the" before "both". Line 20: delete "relatively". 19. Page 16, line 7: Suggest to replace "Two deficiencies are observed in the 3DVar analysis" by "Two problems of RadZIceVar were found in our test".

―――――――――――――――――――――

---

## Short Comment (SC1) · 24 May 2019

I am writing as an executive editor of GMD to highlight issues with the code availability section which needs to be remedied in the revised manuscript.

[Figure]

**Code availability**

The model code is absent. WRF is an open source model, so the source code for the exact version of WRFDA presented here can and must be publicly archived for the reference of future readers. Many GMD authors find Zenodo a suitable archive (https://zenodo.org). I note that WRF is currently developed on GitHub. GitHub is an excellent development platform, and can be referred to from the code availability section as a preferred download location, but it is not a suitable archive location. A citation of an archived version must be included in the manuscript. This is a hard requirement and the manuscript should be rejected if this is not done.

**Data availability**

The data used and generated in this manuscript is only available "on request". This us not usually acceptable in GMD (it would only be acceptable if there were no way for this data to be made available). Please ensure that all of the input data, configuration files, and pre- and post-processing scripts required are persistently and publicly archived, and cited from the code and data availability section. The guiding principle here is that a reader should be able to reproduce the paper using only the information referenced. This is also a strong requirement on the manuscript.

---

## Referee Comment (RC2) · Anonymous Referee #2 · 7 Jun 2019

This paper presents a forward operator, tangent linear model and its adjoint for reflectivity data assimilation (DA) in the variational framework. The procedures are very well documented in details, and the new systems are thoroughly evaluated. This manuscript could serve as a guideline for those who wish to develop tangent linear and adjoint models for new observation types. Although I believe this paper will make a valuable contribution to GMD, I have some concerns about the performance of the forward operator that need to be addressed. I have outlined my concerns below, and my overall recommendation is for acceptance pending major revisions.

1. Some sentences are a bit confusing; for example, "A fixed N0r value is only available for a single-moment microphysics scheme." This could be rephrased to something like, "Nor values are typically fixed (or constant) in single-moment microphysics schemes." Other sentences could be revised to improve readability. 2. Tables 1-2: The coefficients shown in Tables 1 and 2 were adopted from J08. In J08, they derived those coefficients for snow and hail. Therefore, the coefficients the authors adopted for graupel in this manuscript are indeed valid for hail and would result in reflectivity that is too high for graupel, which has a lower density than hail. Those coefficients should be replaced with coefficients for graupel. 3. Fig. 1b: It is hard to understand why the reflectivity for snow is much higher than the reflectivity for graupel. Because snow has a significantly lower density than graupel and the authors adopted hail coefficients for graupel, the reflectivity for graupel should be higher than the reflectivity for snow. Fig. 2 of J08 shows that backscattering amplitudes of hail as a function of the water fraction are larger than those of snow, which is opposite to Fig. 1b here. 4. Page 4, eq (5): Which value is used for the density of graupel? Is it 500 kg/m3 (typical value for graupel) or  $\sim$ 913 kg/m3 (typical value for hail)? 5. Page 10, line 10: Why is the Thompson microphysics scheme used in this study? There are big differences between the snow and graupel size spectra assumptions used in the Thompson microphysics scheme and used in their DA system. This mismatch likely requires significant internal adjustments among state variables when the forecast model is launched after DA. For the purpose of evaluating the performance of their new system, it would make more sense to use a single-moment microphysics scheme that is consistent with their radar DA system. 6. Page 11, line 10: One dBZ observation error is too small, even if the performance does not change significantly with larger observation errors. 7. Page 11, line 19: The authors may use root-mean-square innovation (RMSI) instead of root-mean-square error (RMSE). 8. Page 12, lines 19-24: The operator implemented in CAPS-PRS is not the operator presented in J08 but the one developed by Jung et al. (2010, JAMC), which uses the numerical integration of the T-matrix scattering amplitudes over the particle
size distribution (PSD). This one includes the Mie effect as well. By default, CAPS-

PRS would use the particle size distributions that are consistent with those used in the Thompson microphysics scheme. This means that the snow PSD is the combination of the gamma and exponential distributions, and the graupel PSD uses a diagnostic intercept parameter instead of a fixed value. Therefore, the almost exact fit between J08orig and RefZlceVar in Fig. 4 surprises me. I wonder if a mistake was made here. 9. Page 13, lines 27-29: However, 0 dBZ observations are available in clear air. Are they not used to suppress spurious echoes? 10. Fig. 14: Please add the line for thresholds for a skillful forecast and climatology.

---

## Author Comment (AC1) · 20 Jul 2019

According to editor's comment, we uploaded the code of RadarVar v1.0 and scripts for running experiments. The RadarVar v1.0 operator is integrated into the community WRFDA software and will be publicly available in a future release. The code of RadarVar v1.0 and the scripts for running experiments in this study can be obtained at https://github.com/children1985/WRFDA_gmd. The GFS data are available at https://www.ncdc.noaa.gov/data-access/model-data/model-datasets/global-forcast-system-gfs and the radar data can be downloaded at

https://www.ncdc.noaa.gov/nexradinv/.
* * *

---

## Author Comment (AC2) · 20 Jul 2019

Please see the supplement (formatted)

Reply to review 1

This paper describes an radar reflectivity operator and its adjoint for variational data assimilation. The new operator is implemented in WRFDA and preliminary test is presented with a convective case occurred in the U.S.. The developmental procedures of the operator is well described and the results are well explained and analyzed. My
recommendation is to publish on GMD after minor revision. Although I do not have any major concerns about the content of the paper, I do suggest that the authors pay good attention to improve the English writing. Below are some suggestions from me to help improve the readability of the paper, but I strongly suggest the authors to hire a professional editor to further improve the paper.

We thank the reviewer for her careful review of our paper, and for her helpful comments that improved our paper. Our responses are given below in bold. RadZIceVar is renamed as RadarVar in the revision.

1. Title: The abbreviated algorithm name is not necessary in the title.

Fixed as reviewer's comment

2. Abstract, lines 15-18: Change to "It is shown that the deficiencies in the analysis using this operator, caused by the poor quality. . ..error covariance, can be partially resolved . . ..".

Fixed as reviewer's comment

3. Page 1, lines 24-25: ". . .Xue et al., 2006) and they have demonstrated that assimilating these observations improves. . ."

Fixed as reviewer's comment

4. Page 2, line 1: Change "limited in" to "limited to".

Fixed as reviewer's comment

5. Page 2, line 12: Add references for this statement.

Reference is added: "Their expressions were derived according to the scattering amplitudes that were estimated through the T-matrix method and the Rayleigh scattering approximation (J08)"

6. Page 2, line 13: Change "and the conditions that" to "in which".

Fixed as reviewer's comment

7. Page 2, line 22: Change this sentence to "Reflectivity operators have been developed both for the variational method (. . ..) and for the ensemble Kalman filter method. . .".

Fixed as reviewer's comment

8. Page 2, line 30: Suggested change: "Despite the difficulty, some efforts have been undertaken for reflectivity assimilation . . .."

Fixed as reviewer's comment

9. Page 3, line 14: "To compute Eq. (2), the mixing ratios of . . ... are required."

Fixed as reviewer's comment

10. Sections 2.1.2 and 2.1.3: There are so many parameters, such as those in Eqs. (11) and (16)-(20). Can you briefly explain the meanings of these parameters? Are they theoretically or empirically determined? What are their uncertainties?

For Eq. (11), most parameters are previously introduced (e.g., $\Gamma$, $\lambda$, $\pi$, |Kw|2) or will be described in the following text. The value of the intercept parameter N0x is given without any explanation, thus we add some words for this parameter (page 5, line 2-4). "The intercept parameters of these species are denoted by N0x, the values of which are $3\times106$ m-4 and $4\times105$ m-4 for snow and graupel, respectively. Both values are consistent with the default values of ARPS EnKF where the J08 operator was implemented."

For Eq. (13), all parameters are duplications of J08. For readers not familiar with J08, we add some explanations for these parameters. For the standard deviation of canting angle, we add a reference to show its impact on differential reflectivity (page 5, line 10-12). "According to J08, is zero for all hydrometeors, and $\sigma$ is assumed to be different for snow ($20°$) and hail ($60°$). Here, we assume that $\sigma$ for graupel is also $60°$. The

horizontal reflectivity that is concerned in this study is not sensitive to the canting angle (will be demonstrated in section 2.4), although the differential reflectivity is sensitive to canting angle (Aydin and Seliga, 1984)." For the backscattering amplitude associated parameters ($\alpha$dxa and $\alpha$dxb), we adopt the comment of reviewer 2 and recalculated these parameters using backscattering amplitude in pyCAPS (page 5, line 14-18). "The coefficients of graupel are calculated using the backscattering amplitudes (for particle size <10 mm) in the pyCAPS-PRS v1.1 software (Dawson et al., 2014; Johnson et al., 2016; Jung et al., 2010; Jung et al., 2008) provided by the Center for Analysis and Prediction of Storms (CAPS), and fitted to the polynomial function of fwg. $\alpha$dga and $\alpha$dgb are the coefficients when fwg is zero which means no rainwater."

For Eqs. (16)-(20), all parameters are based on J08 except for the coefficients of graupel. Some variables are intermediate variables that are used to simplify the expression of J08. The derivation of these intermediate variables is shown in appendix. To make our statement clearer, we add some words in the related sentences (page 6, line 8-10). "where "x" is "s" (g) for snow (graupel), $\varepsilon$x is 10-4 (10-3) for snow (graupel), Pwxak and Pwxbk are precalculated constants for S-band radar, the value of n is 6, and the superscript k denotes the index of these constants. All these values are based on J08 except for those of graupel which are computed using the same method mentioned in 2.1.2." (page 7, line 2) "where Pwxai and Pwxbi are precalculated constants in Eq. (16) and are listed in Table 2."

11. Page 9, line 14: ". . .more substantially than that from dry snow. . ."

Fixed as reviewer's comment

12. Page 9, line 24: ". . ..from southern South Dakota to northern Nebraska, as shown in Fig.2. Note that there is also a weaker precipitation system near the north boundary of the domain. The top of the convective system of interest at this time, identified by reflectivity greater than 5 dBZ, reached 16 AGL". In the next line, "a bow echo was observed. . .". Line 3: delete "as shown in Fig. 2" here.

Fixed as reviewer's comment

13. Page 10, line 25: Change "that of snow and graupel" to "those of snow and graupel". Line 27: do you mean a broad vertical distribution?

Yes, a broad vertical distribution.

14. Page 11, line 25: "More outer loops were necessary due to the inaccurate . . .". Lines 26-28: do you mean a total of four experiments were performed by varying the number of iterations and the analysis time?

The experiments are Exp_ref and two of its variants. To make the statement clearer, we modify the related sentence to (page 12, line 17-19): "To determine the tradeoff between the analysis quality and computational cost, two variants of Exp_ref were conducted with 50 and 100 inner iterations. In each experiment, the radar DA analyses were performed at 00Z and 01Z."

15. Page 11, line 30: "RadZIceVar is unable to create hydrometeor increments. . ."?

Revised as "Note that TL/AD of RadarVar will not be able to create reflectivity increments with the zero-hydrometeor background"

16. Page 12, line 2: ". . .constant, it is expected that. . .". This should be a general expectation, so you do not have to refer to J08. Line 7: "nonzero" instead of "nonvanishing". Line 9: "To examine the analysis performance. . .". Line 14: add "the" before "length scale".

Fixed as reviewer's comment

17. Page 13, line 16: Why was such a small weighting 0.1 used? Did you tried any other weighting?

We have tried other weighting. In the case of using a larger weight coefficient, the impact of RadarVar will be weakened because the difference between the background and the observation becomes small. The initial cost function value with weighting 1.0

is 10 times smaller than that with weighting 0.1.

To make the statement clear, we add the following sentences (page 12, line 31 – page 13, line 1) "In addition, current RadarVar cannot work with the weight coefficient of the retrieval part being smaller than $6\times10$-4 if the background contains no hydrometeor. A larger weight coefficient of the retrieval part (e.g., 0.5) reduces the difference between the background and the observation, which weakens the impact of direct DA using RadarVar and is contradictory to the purpose of this study."

18. Page 13, line 17: The use of "reflectivity space" and "model space" are not appropriate in this context. Also in line 5 on page 15. Line 18: delete "the" before "both". Line 20: delete "relatively".

The related words are is rewritten as "in terms of the radar reflectivity and the mixing ratios of rain, snow, and graupel" (page 14, line 12-13) and "in terms of the radar reflectivity and hydrometeor mixing ratios" (page 16, line 6)

19. Page 16, line 7: Suggest to replace "Two deficiencies are observed in the 3DVar analysis" by "Two problems of RadarVar were found in our test".

Fixed as reviewer's comment

 

For Review #2

This paper presents a forward operator, tangent linear model and its adjoint for reflectivity data assimilation (DA) in the variational framework. The procedures are very well documented in details, and the new systems are thoroughly evaluated. This manuscript could serve as a guideline for those who wish to develop tangent linear and adjoint models for new observation types. Although I believe this paper will make a valuable contribution to GMD, I have some concerns about the performance of the forward operator that need to be addressed. I have outlined my concerns below, and my overall recommendation is for acceptance pending major revisions.

We thank the reviewer for his/her careful review of our paper, and for his/her helpful comments that improved our paper. Our responses are given below in bold. RadZIce-Var is renamed as RadarVar in the revision.

1. Some sentences are a bit confusing; for example, "A fixed N0r value is only available for a single-moment microphysics scheme." This could be rephrased to something like, "N0r values are typically fixed (or constant) in single-moment microphysics schemes." Other sentences could be revised to improve readability.

Fixed as reviewer's comments

2. Tables 1-2: The coefficients shown in Tables 1 and 2 were adopted from J08. In J08, they derived those coefficients for snow and hail. Therefore, the coefficients the authors adopted for graupel in this manuscript are indeed valid for hail and would result in reflectivity that is too high for graupel, which has a lower density than hail. Those coefficients should be replaced with coefficients for graupel.

Thanks for pointing out this.

Authors acknowledge that it is more proper to use graupel coefficients in this study. Therefore, we computed the graupel coefficients, $f_a(\pi)$ and $f_b(\pi)$, in the revision. These coefficients were computed using the backscatter amplitudes of graupel stored in py-CAPS software. In J08, the Rayleigh assumption was adopted for simplifying the equation and reducing the computational cost, but in pyCAPS, the backscatter amplitudes are computed using the T-matrix method and does not fully fit the Rayleigh assumption. However, for particle size smaller than 12 mm, the |fa|, as well as |fb| follows the relationship $|f_a|=\alpha_{xa}D^{\beta_{xa}}$ with a fixed $\alpha_{xa}$ and $\beta_{xa}$ (=3.0) which is consistent with the assumption made in J08. This relationship is still approximately valid until the particle size is greater than 22 mm. J08 mentioned that the Rayleigh assumption resulted in reflectivity being overestimated for large particles; thus this drawback naturally exists in RadarVar. The following figure shows |fa| and |fb| as functions of fw (water fraction).

The backscatter amplitudes of hail are shown with gray solid lines; they are higher than their graupel counterparts (red for fa and blue for fb) as reviewer mentioned. Circles are fitting results using our graupel coefficients. These new coefficients are listed in Tables 1 and 2 in the revision.

All results except for those in section 4 (to be compared to the original J08 operator) are recalculated using the above graupel coefficients. The new results (scatter plots, cost functions, and the analysis reflectivity patterns) look similar to the previous results using hail coefficient except for some details.

3. Fig. 1b: It is hard to understand why the reflectivity for snow is much higher than the reflectivity for graupel. Because snow has a significantly lower density than graupel and the authors adopted hail coefficients for graupel, the reflectivity for graupel should be higher than the reflectivity for snow. Fig. 2 of J08 shows that backscattering amplitudes of hail as a function of the water fraction are larger than those of snow, which is opposite to Fig. 1b here.

It was our mistake when plotting this figure that wrong coefficients of hail phase (10 times smaller than the correct values) were used. Data in this plot has been updated in the revision.

4. Page 4, eq (5): Which value is used for the density of graupel? Is it 500 kg/m3 (typical value for graupel) or 913 kg/m3 (typical value for hail)?

The density of 400 kg m-3 was used. Meanwhile, the intercept parameter was also set to the value of graupel ($4\times10^5$). Both values are obtained from ARPS EnKF where the original J08 operator was implemented.

To clarify the density value, we add "where x is either the density of snow (100 kg m-3) or graupel (400 kg m-3)" at page 5, line 7.

5. Page 10, line 10: Why is the Thompson microphysics scheme used in this study? There are big differences between the snow and graupel size spectra assumptions

used in the Thompson microphysics scheme and used in their DA system. This mismatch likely requires significant internal adjustments among state variables when the forecast model is launched after DA. For the purpose of evaluating the performance of their new system, it would make more sense to use a single-moment microphysics scheme that is consistent with their radar DA system.

Authors acknowledge that it would be better to use consistent MP parameters between the reflectivity operator in DA system and the microphysics scheme in model forecast. However, there is no improvement in terms of the forecast skill (FSS) and the initial RMSI difference between two cycles (00Z and 01Z) using a single-moment microphysics scheme (Goddard) with the density and intercept parameters being identical to RadarVar. For example, the initial RMSIs at 00Z and 01Z are $1.6 \times 106$ and $1.3 \times 106$ when Thompson scheme is used, while they are about $1.6 \times 106$ when the single moment scheme is used. With respect to the forecast skill, there is no improvement when the single-moment scheme is used. Generally, it is not uncommon that DA system could use different assumptions from model. For example, scattering radiative transfer model for cloudy radiance DA does not always assume the same particle size distribution as WRF model's MP schemes. WRF has a lot of MP schemes, this radar operator was not designed specifically for Thompson scheme and we just use the Thompson scheme for the tests.

To make our statement clearer, we added the following sentence (page 10, line 21 – page 11, line 4): "Note that in current RadarVar implementation, the intercept parameters are fixed, while they spatiotemporally vary in the Thompson scheme. This inconsistent may increase the adjustment time for model initialization. However, this issue is secondary in the present because no improvement in terms of forecast skill which will be introduced was found in our early tests using a single-moment microphysics scheme with the density and intercept parameters being identical to RadarVar. The primary DA issue in this study is the poor background quality (due to no hydrometeor in GFS analysis or the precipitation displacement). The inconsistent between the

operator and microphysics scheme will be considered in the future"

6. Page 11, line 10: One dBZ observation error is too small, even if the performance does not change significantly with larger observation errors.

We reset the error to 2 dBZ, being consistent with J08 paper, and recalculated all results using this new observation error.

7. Page 11, line 19: The authors may use root-mean-square innovation (RMSI) instead of root-mean-square error (RMSE).

In data assimilation terminology, "innovation" is specifically referred to the background departure to observations and here we also talked about the analysis departure. So we changed to root-mean-square difference (RMSD).

8. Page 12, lines 19-24: The operator implemented in CAPS-PRS is not the operator presented in J08 but the one developed by Jung et al. (2010, JAMC), which uses the numerical integration of the T-matrix scattering amplitudes over the particle size distribution (PSD). This one includes the Mie effect as well. By default, CAPS-PRS would use the particle size distributions that are consistent with those used in the Thompson microphysics scheme. This means that the snow PSD is the combination of the gamma and exponential distributions, and the graupel PSD uses a diagnostic intercept parameter instead of a fixed value. Therefore, the almost exact fit between J08orig and RefZIceVar in Fig. 4 surprises me. I wonder if a mistake was made here.

Thanks for pointing this out. Authors made a mistake in the statement; the results of the original J08 operator actually came from ARPS EnKF. We corrected the statement in the revision.

9. Page 13, lines 27-29: However, 0 dBZ observations are available in clear air. Are they not used to suppress spurious echoes?

Thanks for reviewer's comment. In current implementation, the observed reflectivity not less than 0 dBZ data were used, while other data were assigned as missing data.

The missing data are sometimes defined as non-precipitation echo, but how to define non-precipitation echo is still an open question. Some of studies define Z<5 dBZ as non-precipitation echo while some of other studies use threshold of -15 dBZ. We plan to add non-precipitation echo in future work. Authors acknowledge this issue and add the following statement to note this issue (page 14, line 27-page 15, line 1). "Another cause of these spurious echoes is that non-precipitation echo was not assigned in the observation data in this study such that DA has no impact outside the observed convective area. Therefore, an approach to suppress the spurious echoes is to determine the non-precipitation points, assign a specific value like 0 dBZ to these points, and assimilate these non-precipitation echoes. The non-precipitation echoes will be considered in the future."

10. Fig. 14: Please add the line for thresholds for a skillful forecast and climatology.

Added as reviewer's comment.

Please also note the supplement to this comment:
https://www.geosci-model-dev-discuss.net/gmd-2019-67/gmd-2019-67-AC2-supplement.pdf
* * *
[Figure]

**Fig. 1.** The backscatter amplitudes of hail (gray lines) and graupel (colored lines) in pyCAPS. The circles denote the polynomial fitting results

---

## Author Response (AR1)

Reply to review 1

This paper describes an radar reflectivity operator and its adjoint for variational data assimilation. The new operator is implemented in WRFDA and preliminary test is presented with a convective case occurred in the U.S.. The developmental procedures of the operator is well described and the results are well explained and analyzed. My recommendation is to publish on GMD after minor revision. Although I do not have any major concerns about the content of the paper, I do suggest that the authors pay good attention to improve the English writing. Below are some suggestions from me to help improve the readability of the paper, but I strongly suggest the authors to hire a professional editor to further improve the paper.

**We thank the reviewer for her careful review of our paper, and for her helpful comments that improved our paper. Our responses are given below in bold. RadZIceVar is renamed as RadarVar in the revision.**

1. Title: The abbreviated algorithm name is not necessary in the title.

**Fixed as reviewer's comment**

2. Abstract, lines 15-18: Change to "It is shown that the deficiencies in the analysis using this operator, caused by the poor quality. . ..error covariance, can be partially resolved . . ..".

**Fixed as reviewer's comment**

3. Page 1, lines 24-25: ". . .Xue et al., 2006) and they have demonstrated that assimilating these observations improves. . ."

**Fixed as reviewer's comment**

4. Page 2, line 1: Change "limited in" to "limited to".

**Fixed as reviewer's comment**

5. Page 2, line 12: Add references for this statement.

**Reference is added: "*Their expressions were derived according to the scattering amplitudes that were estimated through the T-matrix method and the Rayleigh scattering approximation (J08)*"**

6. Page 2, line 13: Change "and the conditions that" to "in which".

**Fixed as reviewer's comment**

7. Page 2, line 22: Change this sentence to "Reflectivity operators have been developed both for the variational method (. . ..) and for the ensemble Kalman filter method. . .".

**Fixed as reviewer's comment**

8. Page 2, line 30: Suggested change: "Despite the difficulty, some efforts have been undertaken for reflectivity assimilation . . .."

**Fixed as reviewer's comment**

9. Page 3, line 14: "To compute Eq. (2), the mixing ratios of . . ... are required."

**Fixed as reviewer's comment**

10. Sections 2.1.2 and 2.1.3: There are so many parameters, such as those in Eqs. (11) and (16)-(20). Can you briefly explain the meanings of these parameters? Are they theoretically or empirically determined? What are their uncertainties?

**For Eq. (11), most parameters are previously introduced (e.g., $\Gamma$, $\lambda$, $\pi$, $|Kw|^2$) or will be described in the following text. The value of the intercept parameter $N_{0x}$ is given without any explanation, thus we add some words for this parameter (page 5, line 2-4).**
*"The intercept parameters of these species are denoted by $N_{0x}$, the values of which are $3 \times 10^6$ $m^{-4}$ and $4 \times 10^5$ $m^{-4}$ for snow and graupel, respectively. Both values are consistent with the default values of ARPS EnKF where the J08 operator was implemented."*

**For Eq. (13), all parameters are duplications of J08. For readers not familiar with J08, we add some explanations for these parameters. For the standard deviation of canting angle, we add a reference to show its impact on differential reflectivity (page 5, line 10-12).**

*"According to J08, $\overline{\phi}$ is zero for all hydrometeors, and $\sigma$ is assumed to be different for snow (20°) and hail (60°). Here, we assume that $\sigma$ for graupel is also 60°. The horizontal reflectivity that is concerned in this study is not sensitive to the canting angle (will be demonstrated in section 2.4), although the differential reflectivity is sensitive to canting angle (Aydin and Seliga, 1984)."*

**For the backscattering amplitude associated parameters ($\alpha_{dxa}$ and $\alpha_{dxb}$), we adopt the comment of reviewer 2 and recalculated these parameters using backscattering amplitude in pyCAPS (page 5, line 14-18).**
*"The coefficients of graupel are calculated using the backscattering amplitudes (for particle size <10 mm) in the pyCAPS-PRS v1.1 software (Dawson et al., 2014; Johnson et al., 2016; Jung et al., 2010; Jung et al., 2008) provided by the Center for Analysis and Prediction of*

*Storms (CAPS), and fitted to the polynomial function of $f_{wg}$. $\alpha_{dga}$ and $\alpha_{dgb}$ are the coefficients when $f_{wg}$ is zero which means no rainwater.*"

**For Eqs. (16)-(20), all parameters are based on J08 except for the coefficients of graupel. Some variables are intermediate variables that are used to simplify the expression of J08. The derivation of these intermediate variables is shown in appendix. To make our statement clearer, we add some words in the related sentences (page 6, line 8-10).**
"*where "x" is "s" (g) for snow (graupel), $\varepsilon_x$ is $10^{-4}$ ($10^{-3}$) for snow (graupel), $P_{wxak}$ and $P_{wxbk}$ are precalculated constants for S-band radar, the value of n is 6, and the superscript k denotes the index of these constants. All these values are based on J08 except for those of graupel which are computed using the same method mentioned in 2.1.2.*"
**(page 7, line 2)** "*where $P_{wxai}$ and $P_{wxbi}$ are precalculated constants in Eq. (16) and are listed in Table 2.*"

11. Page 9, line 14: ". . .more substantially than that from dry snow. . ."

**Fixed as reviewer's comment**

12. Page 9, line 24: ". . ..from southern South Dakota to northern Nebraska, as shown in Fig.2. Note that there is also a weaker precipitation system near the north boundary of the domain. The top of the convective system of interest at this time, identified by reflectivity greater than 5 dBZ, reached 16 AGL". In the next line, "a bow echo was observed. . .". Line 3: delete "as shown in Fig. 2" here.

**Fixed as reviewer's comment**

13. Page 10, line 25: Change "that of snow and graupel" to "those of snow and graupel". Line 27: do you mean a broad vertical distribution?

**Yes, a broad vertical distribution.**

14. Page 11, line 25: "More outer loops were necessary due to the inaccurate . . .". Lines 26-28: do you mean a total of four experiments were performed by varying the number of iterations and the analysis time?

**The experiments are Exp_ref and two of its variants. To make the statement clearer, we modify the related sentence to (page 12, line 17-19):**
"*To determine the tradeoff between the analysis quality and computational cost, two variants of Exp_ref were conducted with 50 and 100 inner iterations. In each experiment, the radar DA analyses were performed at 00Z and 01Z.*"

15. Page 11, line 30: "RadZIceVar is unable to create hydrometeor increments. . ."?

**Revised as** "*Note that TL/AD of RadarVar will not be able to create reflectivity increments with the zero-hydrometeor background*"

16. Page 12, line 2: ". . .constant, it is expected that. . .". This should be a general expectation, so you do not have to refer to J08. Line 7: "nonzero" instead of "nonvanishing". Line 9: "To examine the analysis performance. . .". Line 14: add "the" before "length scale".

**Fixed as reviewer's comment**

17. Page 13, line 16: Why was such a small weighting 0.1 used? Did you tried any other weighting?

**We have tried other weighting. In the case of using a larger weight coefficient, the impact of RadarVar will be weakened because the difference between the background and the observation becomes small. The initial cost function value with weighting 1.0 is 10 times smaller than that with weighting 0.1.**

**To make the statement clear, we add the following sentences**
**(page 12, line 31 – page 13, line 1)**
**"*In addition, current RadarVar cannot work with the weight coefficient of the retrieval part being smaller than $6 \times 10^{-4}$ if the background contains no hydrometeor. A larger weight coefficient of the retrieval part (e.g., 0.5) reduces the difference between the background and the observation, which weakens the impact of direct DA using RadarVar and is contradictory to the purpose of this study.*"**

18. Page 13, line 17: The use of "reflectivity space" and "model space" are not appropriate in this context. Also in line 5 on page 15. Line 18: delete "the" before "both". Line 20: delete "relatively".

**The related words are is rewritten as "*in terms of the radar reflectivity and the mixing ratios of rain, snow, and graupel*" (page 14, line 12-13) and "*in terms of the radar reflectivity and hydrometeor mixing ratios*" (page 16, line 6)**

19. Page 16, line 7: Suggest to replace "Two deficiencies are observed in the 3DVar analysis" by "Two problems of RadarVar were found in our test".

**Fixed as reviewer's comment**

For Review #2

This paper presents a forward operator, tangent linear model and its adjoint for reflectivity data assimilation (DA) in the variational framework. The procedures are very well documented in details, and the new systems are thoroughly evaluated. This manuscript could serve as a guideline for those who wish to develop tangent linear and adjoint models for new observation types. Although I believe this paper will make a valuable contribution to GMD, I have some concerns about the performance of the forward operator that need to be addressed. I have outlined my concerns below, and my overall recommendation is for acceptance pending major revisions.

**We thank the reviewer for his/her careful review of our paper, and for his/her helpful comments that improved our paper. Our responses are given below in bold. RadZIceVar is renamed as RadarVar in the revision.**

1. Some sentences are a bit confusing; for example, "A fixed N0r value is only available for a single-moment microphysics scheme." This could be rephrased to something like, "N0r values are typically fixed (or constant) in single-moment microphysics schemes." Other sentences could be revised to improve readability.

**Fixed as reviewer's comments**

2. Tables 1-2: The coefficients shown in Tables 1 and 2 were adopted from J08. In J08, they derived those coefficients for snow and hail. Therefore, the coefficients the authors adopted for graupel in this manuscript are indeed valid for hail and would result in reflectivity that is too high for graupel, which has a lower density than hail. Those coefficients should be replaced with coefficients for graupel.

**Thanks for pointing out this.**

**Authors acknowledge that it is more proper to use graupel coefficients in this study. Therefore, we computed the graupel coefficients, $fa(\pi)$ and $fb(\pi)$, in the revision. These coefficients were computed using the backscatter amplitudes of graupel stored in pyCAPS software. In J08, the Rayleigh assumption was adopted for simplifying the equation and reducing the computational cost, but in pyCAPS, the backscatter amplitudes are computed using the T-matrix method and does not fully fit the Rayleigh assumption. However, for particle size smaller than 12 mm, the $|fa|$, as well as $|fb|$ follows the relationship $|fa|=\alpha_{xa}D^{\beta xa}$ with a fixed $\alpha_{xa}$ and $\beta_{xa}$ (=3.0) which is consistent with the assumption made in J08. This relationship is still approximately valid until the particle size is greater than 22 mm. J08 mentioned that the Rayleigh assumption resulted in reflectivity being overestimated for large particles; thus this drawback naturally exists in RadarVar. The following figure shows $|fa|$ and $|fb|$ as functions of $f_w$ (water fraction).**

[Figure]

The backscatter amplitudes of hail are shown with gray solid lines; they are higher than their graupel counterparts (red for fa and blue for fb) as reviewer mentioned. Circles are fitting results using our graupel coefficients. These new coefficients are listed in Tables 1 and 2 in the revision.

All results except for those in section 4 (to be compared to the original J08 operator) are recalculated using the above graupel coefficients. The new results (scatter plots, cost functions, and the analysis reflectivity patterns) look similar to the previous results using hail coefficient except for some details.

3. Fig. 1b: It is hard to understand why the reflectivity for snow is much higher than the reflectivity for graupel. Because snow has a significantly lower density than graupel and the authors adopted hail coefficients for graupel, the reflectivity for graupel should be higher than the reflectivity for snow. Fig. 2 of J08 shows that backscattering amplitudes of hail as a function of the water fraction are larger than those of snow, which is opposite to Fig. 1b here.

It was our mistake when plotting this figure that wrong coefficients of hail phase (10 times smaller than the correct values) were used. Data in this plot has been updated in the revision.

4. Page 4, eq (5): Which value is used for the density of graupel? Is it 500 kg/m3 (typical value for graupel) or 913 kg/m3 (typical value for hail)?

The density of 400 kg m$^{-3}$ was used. Meanwhile, the intercept parameter was also set to the value of graupel ($4 \times 10^5$). Both values are obtained from ARPS EnKF where the original J08 operator was implemented.

To clarify the density value, we add *"where $\rho_x$ is either the density of snow (100 kg m$^{-3}$) or graupel (400 kg m$^{-3}$)"* at page 5, line 7.

5. Page 10, line 10: Why is the Thompson microphysics scheme used in this study? There are big differences between the snow and graupel size spectra assumptions used in the Thompson microphysics scheme and used in their DA system. This mismatch likely requires significant internal adjustments among state variables when the forecast model is launched after DA. For the purpose of evaluating the performance of their new system, it would make more sense to use a single-moment microphysics scheme that is consistent with their radar DA system.

**Authors acknowledge that it would be better to use consistent MP parameters between the reflectivity operator in DA system and the microphysics scheme in model forecast. However, there is no improvement in terms of the forecast skill (FSS) and the initial RMSI difference between two cycles (00Z and 01Z) using a single-moment microphysics scheme (Goddard) with the density and intercept parameters being identical to RadarVar. For example, the initial RMSIs at 00Z and 01Z are $1.6 \times 10^6$ and $1.3 \times 10^6$ when Thompson scheme is used, while they are about $1.6 \times 10^6$ when the single moment scheme is used. With respect to the forecast skill, there is no improvement when the single-moment scheme is used.**
**Generally, it is not uncommon that DA system could use different assumptions from model. For example, scattering radiative transfer model for cloudy radiance DA does not always assume the same particle size distribution as WRF model's MP schemes. WRF has a lot of MP schemes, this radar operator was not designed specifically for Thompson scheme and we just use the Thompson scheme for the tests.**

**To make our statement clearer, we added the following sentence (page 10, line 21 – page 11, line 4):**
**"*Note that in current RadarVar implementation, the intercept parameters are fixed, while they spatiotemporally vary in the Thompson scheme. This inconsistent may increase the adjustment time for model initialization. However, this issue is secondary in the present because no improvement in terms of forecast skill which will be introduced was found in our early tests using a single-moment microphysics scheme with the density and intercept parameters being identical to RadarVar. The primary DA issue in this study is the poor background quality (due to no hydrometeor in GFS analysis or the precipitation displacement). The inconsistent between the operator and microphysics scheme will be considered in the future*"**

6. Page 11, line 10: One dBZ observation error is too small, even if the performance does not change significantly with larger observation errors.

**We reset the error to 2 dBZ, being consistent with J08 paper, and recalculated all results using this new observation error.**

7. Page 11, line 19: The authors may use root-mean-square innovation (RMSI) instead of root-mean-square error (RMSE).

**In data assimilation terminology, "innovation" is specifically referred to the background departure to observations and here we also talked about the analysis departure. So we changed to root-mean-square difference (RMSD).**

8. Page 12, lines 19-24: The operator implemented in CAPS-PRS is not the operator presented in J08 but the one developed by Jung et al. (2010, JAMC), which uses the numerical integration of the T-matrix scattering amplitudes over the particle size distribution (PSD). This one includes the Mie effect as well. By default, CAPS-PRS would use the particle size distributions that are consistent with those used in the Thompson microphysics scheme. This means that the snow PSD is the combination of the gamma and exponential distributions, and the graupel PSD uses a diagnostic intercept parameter instead of a fixed value. Therefore, the almost exact fit between J08orig and RefZIceVar in Fig. 4 surprises me. I wonder if a mistake was made here.

**Thanks for pointing this out.**
**Authors made a mistake in the statement; the results of the original J08 operator actually came from ARPS EnKF. We corrected the statement in the revision.**

9. Page 13, lines 27-29: However, 0 dBZ observations are available in clear air. Are they not used to suppress spurious echoes?

**Thanks for reviewer's comment.**
**In current implementation, the observed reflectivity not less than 0 dBZ data were used, while other data were assigned as missing data. The missing data are sometimes defined as non-precipitation echo, but how to define non-precipitation echo is still an open question. Some of studies define Z<5 dBZ as non-precipitation echo while some of other studies use threshold of -15 dBZ. We plan to add non-precipitation echo in future work.**
**Authors acknowledge this issue and add the following statement to note this issue (page 14, line 27-page 15, line 1).**
**"*Another cause of these spurious echoes is that non-precipitation echo was not assigned in the observation data in this study such that DA has no impact outside the observed convective area. Therefore, an approach to suppress the spurious echoes is to determine the non-precipitation points, assign a specific value like 0 dBZ to these points, and assimilate these non-precipitation echoes. The non-precipitation echoes will be considered in the future.*"**

10. Fig. 14: Please add the line for thresholds for a skillful forecast and climatology.

**Added as reviewer's comment.**

[revised manuscript text omitted]

$$\sum_{i=0}^{n}[P_{\mathrm{wxa}i}(\sum_{j=0}^{n}P_{\mathrm{wxb}j}f_{\mathrm{wx}}^{i+j})]=\ P_{\mathrm{wxa}0}P_{\mathrm{wxb}0}f_{\mathrm{wx}}^{0}$$

$$+P_{\mathrm{wxa}0}P_{\mathrm{wxb}1}f_{\mathrm{wx}}^{1}+P_{\mathrm{wxa}1}P_{\mathrm{wxb}0}f_{\mathrm{wx}}^{1}$$

$$\dots$$

$$+P_{\mathrm{wxa}0}P_{\mathrm{wxb}n}f_{\mathrm{wx}}^{n}+P_{\mathrm{wxa}1}P_{\mathrm{wxb}(n-1)}f_{\mathrm{wx}}^{n}+\cdots+P_{\mathrm{wxa}n}P_{\mathrm{wxb}0}f_{\mathrm{wx}}^{n}$$

$$+P_{\mathrm{wxa}1}P_{\mathrm{wxb}n}f_{\mathrm{wx}}^{n+1}+P_{\mathrm{wxa}2}P_{\mathrm{wxb}(n-1)}f_{\mathrm{wx}}^{n+1}+\cdots+P_{\mathrm{wxa}n}P_{\mathrm{wxb}1}f_{\mathrm{wx}}^{n+1}$$

$$\dots$$

$$+P_{\mathrm{wxa}(n-1)}P_{\mathrm{wxb}n}f_{\mathrm{wx}}^{2n-1}+P_{\mathrm{wxa}n}P_{\mathrm{wxb}(n-1)}f_{\mathrm{wx}}^{2n-1}$$

$$+P_{\mathrm{wxa}n}P_{\mathrm{wxb}n}f_{\mathrm{wx}}^{2n}$$

$$=\sum_{k=0}^{n}[f_{\mathrm{wx}}^{k}\sum_{i=0}^{k}P_{\mathrm{wxa}i}P_{\mathrm{wxb}(k-i)}]+\sum_{k=n+1}^{2n}[f_{\mathrm{wx}}^{k}\sum_{i=k-n}^{n}P_{\mathrm{wxa}i}P_{\mathrm{wxb}(k-i)}] \qquad \text{A. 3}$$

The sum functions in the square brackets on the right-hand side of A. (3) correspond to the third expression of Eq. (20). Using the third expression of Eq. (20), the third term of A. (2) is rewritten as follows,

$$2C\sum_{i=0}^{n}[P_{\mathrm{wxa}i}(\sum_{j=0}^{n}P_{\mathrm{wxb}j}f_{\mathrm{wx}}^{i+j})]=\ 2C\{\sum_{k=0}^{n}[f_{\mathrm{wx}}^{k}\sum_{i=0}^{k}P_{\mathrm{wxa}i}P_{\mathrm{wxb}(k-i)}]+\sum_{k=n+1}^{2n}[f_{\mathrm{wx}}^{k}\sum_{i=k-n}^{n}P_{\mathrm{wxa}i}P_{\mathrm{wxb}(k-i)}]\}$$

$$=2C\sum_{k=0}^{2n}P_{\mathrm{C}xk}f_{\mathrm{wx}}^{k} \qquad , \qquad \text{A. 4}$$

where $P_{Cxk}$ has the same meaning as in Eq. (20). Similarly, we can rewrite the other two terms in A. (2) as follows:

$$A\sum_{i=0}^{n}[P_{\mathrm{wxa}i}(\sum_{j=0}^{n}P_{\mathrm{wxa}j}f_{\mathrm{wx}}^{i+j})]=A\{\sum_{k=0}^{n}[f_{\mathrm{wx}}^{k}\sum_{i=0}^{k}P_{\mathrm{wxa}i}P_{\mathrm{wxa}(k-i)}]+\sum_{k=n+1}^{2n}[f_{\mathrm{wx}}^{k}\sum_{i=k-n}^{n}P_{\mathrm{wxa}i}P_{\mathrm{wxa}(k-i)}]\}=A\sum_{k=0}^{2n}P_{\mathrm{A}xk}f_{\mathrm{wx}}^{k} \qquad \text{A. 5}$$

$$B\sum_{i=0}^{n}[P_{\mathrm{wxb}i}(\sum_{j=0}^{n}P_{\mathrm{wxb}j}f_{\mathrm{wx}}^{i+j})]=B\{\sum_{k=0}^{n}[f_{\mathrm{wx}}^{k}\sum_{i=0}^{k}P_{\mathrm{wxb}i}P_{\mathrm{wxb}(k-i)}]+\sum_{k=n+1}^{2n}[f_{\mathrm{wx}}^{k}\sum_{i=k-n}^{n}P_{\mathrm{wxb}i}P_{\mathrm{wxb}(k-i)}]\}=B\sum_{k=0}^{2n}P_{\mathrm{B}xk}f_{\mathrm{wx}}^{k}$$

Because these three expressions (A. (4) and A. (5)) contain the same sum function with respect to $k$ from 0 to $2n$, A. (2) can be rewritten as follows:

$$G(f_{\mathrm{wx}})=\varepsilon_{\mathrm{x}}^{2}(A\sum_{k=0}^{2n}P_{\mathrm{A}xk}f_{\mathrm{wx}}^{k}+B\sum_{k=0}^{2n}P_{\mathrm{B}xk}f_{\mathrm{wx}}^{k}+2C\sum_{k=0}^{2n}P_{\mathrm{C}xk}f_{\mathrm{wx}}^{k})$$

$$=\varepsilon_{\mathrm{x}}^{2}\sum_{k=0}^{2n}(AP_{\mathrm{A}xk}f_{\mathrm{wx}}^{k}+BP_{\mathrm{B}xk}f_{\mathrm{wx}}^{k}+2CP_{\mathrm{C}xk}f_{\mathrm{wx}}^{k}) \qquad . \qquad \text{A. 6}$$

$$=\varepsilon_{\mathrm{x}}^{2}\sum_{k=0}^{2n}(AP_{\mathrm{A}xk}+BP_{\mathrm{B}xk}+2CP_{\mathrm{C}xk})f_{\mathrm{wx}}^{k}$$

[revised manuscript text omitted]

 max: 4.7×10$^{-4}$ | min: -4.2×10$^{-4}$
 max: 4.4×10$^{-4}$ | min: -3.7×10$^{-4}$
 max: 3.1×10$^{-4}$ | 1.00114799 |
| sample 2 | min: -1.6×10$^{-6}$
 max: 1.8×10$^{-6}$ | min: -1.6×10$^{-6}$
 max: 1.7×10$^{-6}$ | min: -1.4×10$^{-6}$
 max: 1.2×10$^{-6}$ | 1.00004709 |

**Figures**

[Figure]

**Fig. 1 (a) The reflectivity as (a) a function of $q_r$ (red), $q_s$ (green), and $q_g$ (blue) for pure water and dry snow/graupel and (b) a function of $q_r$–$q_s$ (colors) and $q_r$–$q_g$ (contours) for wet snow/graupel.**

[Figure]

**Fig. 2 The simulation domain with radar sites marked by radar icons and names. The areas of precipitation greater than 5 mm h$^{-1}$ are plotted for 00Z (red), 02Z (green), and 04Z (blue) on 2 June 2018.**

[Figure]

**Fig. 3 (a-c) The standard deviations of the background errors at different vertical levels, (d-f) the horizontal correlation length scale as a function of EOF mode, and (g-i) the vertical correlation coefficients for $q_r$, $q_s$, and $q_g$.**

[Figure]

**Fig. 4 Scatter plot of the reflectivity for J08orig (*x* axis) versus RadarVar (*y* axis). The bias, standard deviation (STD), and number of samples are listed in the plot.**

[Figure]

Fig. 5 (a) Vertical distributions of the maximum absolute values of the perturbed $q_r$ (red), $q_s$ (green), and $q_g$ (blue). Reflectivity scatter plots of all_prt (*x* axis) versus (b) F_unprt (*y* axis), (c) rhom_unprt (*y* axis), and (d) SD_unprt (*y* axis) at model levels 1~12 (black, lower than 3 km AGL), 12~20 (red, between 3~7 km AGL), and 21~30 (green, above 7 km AGL).

[Figure]

**Fig. 6 Scatter plots of the observed (*x* axis) versus (a, c) background and (b, d) analysis reflectivity at (a, b) 00Z and (c, d) 01Z in Exp_ref. The mean bias and standard deviation (STD) between the observations and the background (or analysis) are listed in each plot.**

[Figure]

**Fig. 7** The (a, e, i, c, g, k) observed (KLNX) and (b, f, j, d, h, l) analysis reflectivities at (a-d) 2 km, (e-h) 4 km, and (i-l) 6 km AGL at (a, e, i, b, f, j) 00Z and (c, g, k, d, h, l) 01Z in Exp_ref.

[Figure]

**Fig. 8** The (a, b, c) 00Z and (d, e, f) 01Z analyses of $q_r$ (red), $q_s$ (green), and $q_g$ (blue) at model levels 5 (left column), 15 (middle column), and 20 (right column) for outer loop 6 in Exp_ref. Model levels 5, 15, and 20 approximately correspond to 0.7 km, 4.0 km, and 8.0 km AGL, respectively.

[Figure]

**Fig. 9 The cost function and normalized gradient norm as functions of the iteration step during the minimization at (a, c) 00Z and (b, d) 01Z in Exp_ref.**

[Figure]

**Fig. 10** The correlation coefficients of $q_r$ (red), $q_s$ (green), $q_g$ (blue), and reflectivity (gray) between the Exp_ref analysis (with six outer loops and 150 inner iterations each loop) and that with different numbers of outer loops and inner iterations at (a) 00Z and (b) 01Z.

[Figure]

**Fig. 11 Same as Fig. 6 but for the experiments without hydrometeor preprocessing at (a, b) 00Z and (c, d) 01Z.**

[Figure]

**Fig. 12 (a, c) The analysis reflectivity at 4 km AGL and (b, d) the analyses of** $q_r$ **(red),** $q_s$ **(green), and** $q_g$ **(blue) at model level 15 at 00Z and 01Z for the experiments without hydrometeor preprocessing.**

[Figure]

**Fig. 13 Same as Fig. 7 but for (a) the observations and the analyses from (b) Exp_ref, (c) Exp_ls0.5, and (d) Exp_ls0.125.**

[Figure]

**Fig. 14 The FSSs of the hourly precipitation forecasts aggregated over the period from 02Z to 05Z as functions of the radius of influence for forecasts launched from the (a) 00Z and (b) 01Z analyses for Exp_ref (solid lines) and noDA (dashed lines). The hourly precipitation thresholds are denoted by green (1 mm h$^{-1}$), blue (5 mm h$^{-1}$), orange (10 mm h$^{-1}$), and red (20 mm h$^{-1}$) lines. The dashed straight line represents the skillful FSS at 1 mm h$^{-1}$.**